# FROM PROBABILITY TO COUNTERFACTUALS: THE INCREASING COMPLEXITY OF SATISFIABILITY IN PEARL'S CAUSAL HIERARCHY

**Julian Dörfler** *
Saarland University, Germany
jdoerfler@cs.uni-saarland.de

**Benito van der Zander** *
University of Lübeck, Germany
b.vanderzander@uni-luebeck.de

**Markus Bläser** †
Saarland University, Germany
mblaeser@cs.uni-saarland.de

**Maciej Liśkiewicz** †
University of Lübeck, Germany
maciej.liskiewicz@uni-luebeck.de

## ABSTRACT

The framework of Pearl's Causal Hierarchy (PCH) formalizes three types of reasoning: probabilistic (i.e. purely observational), interventional, and counterfactual, that reflect the progressive sophistication of human thought regarding causation. We investigate the computational complexity aspects of reasoning in this framework focusing mainly on satisfiability problems expressed in probabilistic and causal languages across the PCH. That is, given a system of formulas in the standard probabilistic and causal languages, does there exist a model satisfying the formulas?

Our main contribution is to prove the exact computational complexities showing that languages allowing addition and marginalization (via the summation operator) yield $\mathsf{NP^{PP}}$-, $\mathsf{PSPACE}$-, and $\mathsf{NEXP}$-complete satisfiability problems, depending on the level of the PCH. These are the first results to demonstrate a strictly increasing complexity across the PCH: from probabilistic to causal and counterfactual reasoning. On the other hand, in the case of full languages, i.e. allowing addition, marginalization, and multiplication, we show that the satisfiability for the counterfactual level remains the same as for the probabilistic and causal levels, solving an open problem in the field.

## 1 INTRODUCTION

The development of the modern causal theory in AI and empirical sciences has greatly benefited from an influential structured approach to inference about causal phenomena, which is based on a reasoning hierarchy named "Ladder of Causation", also often referred to as the "Pearl's Causal Hierarchy" (PCH) ((Shpitser & Pearl, 2008; Pearl, 2009; Bareinboim et al., 2022), see also (Pearl & Mackenzie, 2018) for a gentle introduction to the topic). This three-level framework formalizes various types of reasoning that reflect the progressive sophistication of human thought regarding causation. It arises from a collection of causal mechanisms that model the "ground truth" of *unobserved* nature formalized within a Structural Causal Model (SCM). These mechanisms are then combined with three patterns of reasoning concerning *observed* phenomena expressed at the corresponding layers of the hierarchy, known as *probabilistic* (also called *associational* in the AI literature), *interventional*, and *counterfactual* (for formal definitions of these concepts, see Sec. 2).

A basic term at the probabilistic/associational layer is expressed as a common probability, such as[1] $\mathbb{P}(x, y)$. This may represent queries like "How likely does a patient have both diabetes $(X = x)$

---

*Contributing equally first authors.

†Contributing equally last authors.

[1]In our paper, we consider random variables over discrete, finite domains. By an event we mean a propositional formula over atomic events of the form $X{=}x$, such as $(X{=}x \wedge Y{=}y)$ or $(X{=}x \vee Y{\neq}y)$. Moreover, by $\mathbb{P}(Y{=}y, X{=}x)$, etc., we mean, as usually, $\mathbb{P}(X{=}x \wedge Y{=}y)$. Finally by $\mathbb{P}(x, y)$, we abbreviate $\mathbb{P}(Y{=}y, X{=}x)$.

and high blood pressure ($Y = y$)?" From basic terms, we can build more complex terms by using additions (linear terms) or even arbitrary polynomials (polynomial terms). This can be combined with the use of a unary summation operator, which allows to express marginalization in a compact way. Formulas at this layer consist of Boolean combinations of (in)equalities of basic, linear, or, in the general case, polynomial terms. The interventional patterns extend the basic probability terms by allowing the use of Pearl's do-operator (Pearl, 2009) which models an experiment like a Randomized Controlled Trial (Fisher, 1936). For instance, $\mathbb{P}([x]y)$ which[2], in general differs from $\mathbb{P}(y|x)$, allows to ask hypothetical questions such as, e.g., "How likely is it that a patient's headache will be cured ($Y = y$) if he or she takes aspirin ($X = x$)?". An example formula at this layer is $\mathbb{P}([x]y) = \sum_z \mathbb{P}(y|x,z)\mathbb{P}(z)$ which estimates the causal effect of the intervention $do(X = x)$ (all patients take aspirin) on outcome variable $Y = y$ (headache cure). It illustrates the use of the prominent back-door adjustment to eliminate the confounding effect of a factor represented by the variable $Z$ (Pearl, 2009). The basic terms at the highest level of the hierarchy enable us to formulate queries related to counterfactual situations. For example, $\mathbb{P}([X{=}x]Y{=}y|(X{=}x', Y{=}y'))$ expresses the probability that, for instance, a patient who did not receive a vaccine ($X = x'$) and died ($Y = y'$) would have lived ($Y = y$) if he or she had been vaccinated ($X = x$).

The computational complexity aspects of reasoning about uncertainty in this framework have been the subject of intensive studies in the past decades, especially in the case of probabilistic inference with the input probability distributions encoded by Bayesian networks (see, e.g., (Pearl, 1988; Cooper, 1990; Dagum & Luby, 1993; Roth, 1996; Park & Darwiche, 2004)). The main focus of our work is on the computational complexity of *satisfiability* problems and their *validity* counterparts which enable formulating precise assumptions on data and implications of causal explanations.

The problems take as input a Boolean combination of (in)equalities of terms at the PCH-layer of interest with the task to decide if there exists a satisfying SCM for the input formula or if the formula is valid for all SCMs, respectively. For example, for binary random variables $X$ and $Y$, the formula consisting of the single equality $\sum_x \sum_y \mathbb{P}((X{=}x) \wedge (X{\neq}x \vee Y{=}y) \wedge (X{\neq}x \vee Y{\neq}y)) = 0$ in the language of the probabilistic layer is satisfied since there exists an SCM in which it is true (in fact, the formula holds in any SCM)[3]. An SCM for inputs at this layer can be identified with the standard joint probability distribution, in our case, with $P(X{=}x, Y{=}y)$, for $x, y \in \{0, 1\}$.

The complexity of the studied satisfiability problems depends on the combination of two factors: (1) the PCH-layer to which the basic terms belong and (2) the operators which can be used to specify the (in)equalities of the input formula. The most basic operators are "+" and "·" (leading to linear, resp., polynomial terms) and, meaningful in causality, the unary summation operator $\Sigma$ used to express marginalization. Of interest is also conditioning, which will be discussed in our paper, as well. The main interest of our research is focused on the precise characterization of the computational complexity of satisfiability problems (and their validity counterparts) for languages of all PCH layers, combined with increasing the expressiveness of (in)equalities by enabling the use of more complex operators.

**Related Work to our Study.** In their seminal paper, Fagin, Halpern, and Megiddo (1990) explore the language of the lowest probabilistic layer of the PCH consisting of Boolean combinations of (in)equalities of *basic* and *linear* terms. Besides the complete axiomatization for the used logic, they show that the problem of deciding satisfiability is NP-complete indicating that the complexity is surprisingly no worse than that of propositional logic. The authors subsequently extend the language to include (in)equalities of *polynomial* terms, aiming to facilitate reasoning about conditional probabilities. While they establish the existence of a PSPACE algorithm for deciding whether such a formula is satisfiable, they leave the exact complexity open. Recently, Mossé, Ibeling, and Icard (2022) resolved this issue by demonstrating that deciding satisfiability is $\exists\mathbb{R}$-complete, where $\exists\mathbb{R}$ represents the well-studied class defined as the closure of the Existential Theory of the Reals (ETR) under polynomial-time many-one reductions. Furthermore, for the higher, more expressive PCH layers Mossé et al. prove that for (in)equalities of polynomial terms both at the interventional and the

---

[2] A common and popular notation for the post-interventional probability is $\mathbb{P}(Y{=}y|do(X{=}x))$. In this paper, however, we use the notation $\mathbb{P}([X{=}x]Y{=}y)$ since it is more convenient for the analysis of counterfactuals.

[3] Interestingly, the instance can be seen as a result of reduction from the not-satisfiable Boolean formula $a \wedge (\overline{a} \vee b) \wedge (\overline{a} \vee \overline{b})$.

counterfactual layer the decision problems still remain $\exists\mathbb{R}$-complete (we recall the definitions of the complexity classes in Sec. 2.2).

The languages used in these studies, and also in other works as, e.g., (Nilsson, 1986; Georgakopoulos et al., 1988; Ibeling & Icard, 2020), are able to fully express probabilistic reasoning, resp., inferring interventional and counterfactual predictions. In particular, they allow one to express *marginalization* which is a common paradigm in this field. However, since the languages *do not* include the unary summation operator $\Sigma$, the abilities of expressing marginalization are relatively limited. Thus, for instance, to express the marginal distribution of a random variable $Y$ over a subset of (binary) variables $\{Z_1, \ldots, Z_m\} \subseteq \{X_1, \ldots, X_n\}$ as $\sum_{z_1,\ldots,z_m} \mathbb{P}(y, z_1, \ldots, z_m)$, an encoding without summation requires an expansion $\mathbb{P}(y, Z_1{=}0, \ldots, Z_m{=}0) + \ldots + \mathbb{P}(y, Z_1{=}1, \ldots, Z_m{=}1)$ of exponential size in $m$. Consequently, to analyze the complexity aspects of the problems under study, languages allowing the standard notation for encoding marginalization using the $\Sigma$ operator are needed. In (van der Zander et al., 2023), the authors present a first systematic study in this setting. They introduce a new natural complexity class, named succ-$\exists\mathbb{R}$, which can be viewed as a succinct variant of $\exists\mathbb{R}$, and show that the satisfiability for the (in)equalities of *polynomial* terms, both at the *probabilistic* and *interventional* layer, are complete for succ-$\exists\mathbb{R}$. They leave open the exact complexity for the *counterfactual* case. Moreover, the remaining variants (basic and linear terms) remain unexplored for all PCH layers.

**Our Contribution.** The previous research establishes that, from a computational perspective, many problems for interventional and counterfactual reasoning are not harder than for pure probabilistic reasoning. In our work, we show that the situation changes significantly if, to express marginalization, the common summation operator is used. Below we highlight our main contributions, partially summarized also in Table 1 which involve complexity classes related to each other as follows[4]:

$$\text{NP} \underset{\subsetneq}{\overset{\subsetneq}{\phantom{x}}} \overset{\exists\mathbb{R}}{\underset{\text{NP}^{\text{PP}}}{\phantom{xx}}} \underset{\subsetneq}{\overset{\subsetneq}{\phantom{x}}} \text{PSPACE} \subseteq \text{NEXP} \subseteq \text{succ-}\exists\mathbb{R} \subseteq \text{EXPSPACE} \tag{1}$$

• For combinations of (in)equalities of *basic* and *linear* terms, unlike previous results, the compact summation for marginalization increases the complexity, depending on the level of the PCH: from $\text{NP}^{\text{PP}}$-, through PSPACE-, to NEXP-completeness.

• The counterfactual satisfiability for (in)equalities of *polynomial* terms is succ-$\exists\mathbb{R}$-complete, which solves the open problem in (van der Zander et al., 2023).

• Accordingly, the validity problems for the languages above are complete for the corresponding complement complexity classes.

| Terms | $\mathcal{L}_1$ (prob.) | $\mathcal{L}_2$ (interv.) | $\mathcal{L}_3$ (count.) |
|---|---|---|---|
| basic lin | NP $(a)$ | | |
| poly | $\exists\mathbb{R}$ $(b)$ | | |
| basic & marg. lin & marg. | $\text{NP}^{\text{PP}}$ (1) | PSPACE (2) | NEXP (3) |
| poly & marg. | succ-$\exists\mathbb{R}$ $(c, 4)$ | | |

Table 1: Completeness results for the satisfiability problems $(a)$ for $\mathcal{L}_1$ (Fagin et al., 1990), for $\mathcal{L}_2$ and $\mathcal{L}_3$ (Mossé et al., 2022), $(b)$ (Mossé et al., 2022), $(c)$ for $\mathcal{L}_1$ and $\mathcal{L}_2$ (van der Zander et al., 2023). Our results (1)-(4): Theorem 4, 8, 9, resp. Theorem 10.

plexity classes. Interestingly, both satisfiability and validity for basic and linear languages with marginalization are PSPACE-complete at the interventional layer.

Our results demonstrate, for the first time, a strictly increasing complexity of reasoning across the PCH – from probabilistic to causal to counterfactual reasoning – under the widely accepted assumption that the inclusions $\text{NP}^{\text{PP}} \subseteq \text{PSPACE} \subseteq \text{NEXP}$ in Eq. (1) above are proper. This relation, in the case of basic and linear languages with marginalization, aligns with the strength of their expressive power: From previous research, we know that the probabilistic languages are less expressive than the causal languages, and the causal languages, in turn, are less expressive than the corresponding counterfactual languages (for more discussion on this, see Sec. 3).

In addition, the impact of establishing exact completeness results for probabilistic, causal, and counterfactual reasoning, as stated in our work, lies in their implications for algorithmic approaches

---

[4]The relationship between $\exists\mathbb{R}$ and $\text{NP}^{\text{PP}}$ is unknown.

to solve these problems. Under widely accepted complexity assumptions like, e.g., NP $\neq$ PSPACE, the completeness of a problem highlights inherent limitations in applying algorithmic techniques, such as dynamic programming, divide-and-conquer, SAT- or ILP-solvers, which are only effective for NP-complete problems. This, in turn, justifies the use of heuristics or algorithms of exponential worst-case complexity. Moreover, using the succ-$\exists\mathbb{R}$-completeness as a yardstick for measuring computational complexity of problems, we show that the complexity of counterfactual reasoning (for the most general queries) remains the same as for common probabilistic reasoning. This is quite a surprising result, as the difference between the expressive power of both settings is huge.

**Structure of the Paper.** In Sec. 2, we provide the main concepts of causation and define formally the problems considered in this work. We derive the complexity of satisfiability for basic and linear languages in Sec. 3 and for polynomial languages in Sec. 4. Due to space constraints, some proofs are omitted from the main text, and only proof outlines are provided. The complete proofs can be found in Sec. A in the appendix. Furthermore, Sec. B in the appendix, provides an example illustrating the three types of reasoning in the framework of the PCH and in Sec. C we give formal definitions for the syntax and semantics of the languages of the hierarchy.

## 2  PRELIMINARIES

In this section, we give definitions of the main concepts of the theory of causation, including the Structural Causal Model (SCM), provide syntax and semantics for the languages of the PCH, and discuss the complexity classes used in the paper. To help understand the formal definitions, we encourage readers unfamiliar with the theory of causation to refer to Section B in the appendix, where we provide an example that, we hope, will make it easier to understand the formal definitions and the intuitions behind them.

### 2.1  THE LANGUAGES OF CAUSAL HIERARCHY

We give here an informal but reasonably precise description of the syntax and semantics of the languages studied in this paper. For formal definitions, see Section C in the appendix.

We always consider discrete distributions and represent the values of the random variables as $Val = \{0, 1, ..., c-1\}$. We denote by $\mathbf{X}$ the set of variables used in a system and by capital letters $X_1, X_2, ...,$ we denote the individual variables. We assume that $Val$ is fixed and of cardinality at least two, and that all variables $X_i$ share the same domain $Val$. A value of $X_i$ is often denoted by $x_i$ or a natural number. By an *atomic* event, we mean an event of the form $X = x$, where $X$ is a random variable and $x$ is a value in the domain of $X$. The language $\mathcal{E}_{prop}$ of propositional formulas $\delta$ over atomic events is defined as the closure of such events under the Boolean operators $\wedge$ and $\neg$. The atomic intervention is either empty $\perp$ or of the form $X = x$. An intervention formula is a conjunction of atomic interventions. The language of post-interventional events, denoted as $\mathcal{E}_{post\text{-}int}$, consists the formulas of the form $[\alpha]\delta$ where $\alpha$ is an intervention and $\delta$ is in $\mathcal{E}_{prop}$. The language of counterfactual events, $\mathcal{E}_{counterfact}$, is the set $\mathcal{E}_{post\text{-}int}$ closed under $\wedge$ and $\neg$.

The PCH consists of languages on three layers each of which is based on terms of the form $\mathbb{P}(\delta_i)$, with $i = 1, 2, 3$. For the observational (associational) language (Layer 1), we have $\delta_1 \in \mathcal{E}_{prop}$, for the interventional language (Layer 2), we have $\delta_2 \in \mathcal{E}_{post\text{-}int}$, and, for the counterfactual language (Layer 3), $\delta_3 \in \mathcal{E}_{counterfact}$. The expressive power and computational complexity properties of the languages depend largely on the operations that are allowed to apply on terms $\mathbb{P}(\delta_i)$. Allowing gradually more complex operators, we define the languages which are the subject of our studies. The terms for levels $i = 1, 2, 3$ are described as follows. The basic terms, denoted as $\mathcal{T}_i^{base}$, are probabilities $\mathbb{P}(\delta_i)$ as, e.g., $\mathbb{P}(X_1{=}x_1 \vee X_2{=}x_2)$ in $\mathcal{T}_1^{base}$ or $\mathbb{P}([X_1{=}x_1]X_2{=}x_2)$ in $\mathcal{T}_2^{base}$. From basic terms, we build more complex linear terms $\mathcal{T}_i^{lin}$ by using additions and polynomial terms $\mathcal{T}_i^{poly}$ by using arbitrary polynomials. By $\mathcal{T}_i^{base\langle\Sigma\rangle}$, $\mathcal{T}_i^{lin\langle\Sigma\rangle}$, and $\mathcal{T}_i^{poly\langle\Sigma\rangle}$, we denote the corresponding sets of terms when including a unary marginalization operator of the form $\sum_x \mathbf{t}$ for a term $\mathbf{t}$. In the summation, we have a dummy variable $x$ which ranges over all values $0, 1, ..., c-1$. The summation $\sum_x \mathbf{t}$ is a purely syntactical concept which represents the sum $\mathbf{t}[^0/_x] + \mathbf{t}[^1/_x] + ... + \mathbf{t}[^{c-1}/_x]$, where by $\mathbf{t}[^v/_x]$, we mean the expression in which all occurrences of $x$ are replaced with

value $v$. For example, for $Val = \{0, 1\}$, the expression $\sum_x \mathbb{P}(Y{=}1, X{=}x)$ semantically represents $\mathbb{P}(Y{=}1, X{=}0) + \mathbb{P}(Y{=}1, X{=}1)$.

Now, let $Lab = \{base, base\langle\Sigma\rangle, lin, lin\langle\Sigma\rangle, poly, poly\langle\Sigma\rangle\}$ denote the labels of all variants of languages. Then for each $* \in Lab$ and $i = 1, 2, 3$, we define the languages $\mathcal{L}_i^*$ of Boolean combinations of inequalities in a standard way by the grammars: $\mathbf{f} ::= \mathbf{t} \le \mathbf{t}' \mid \neg\mathbf{f} \mid \mathbf{f} \wedge \mathbf{f}$ where $\mathbf{t}, \mathbf{t}'$ are terms in $\mathcal{T}_i^*$. Although the languages and their operations can appear rather restricted, all the usual elements of probabilistic and causal formulas can be encoded in a natural way (see Section C in the appendix for more discussion).

To define the semantics, we use SCMs as in (Pearl, 2009, Sec. 3.2). An SCM is a tuple $\mathfrak{M} = (\mathcal{F}, P, \mathbf{U}, \mathbf{X})$, with exogenous variables $\mathbf{U}$ and endogenous variables $\mathbf{X} = \{X_1, \ldots, X_n\}$. $\mathcal{F} = \{F_1, \ldots, F_n\}$ consists of functions such that $F_i$ calculates the value of variable $X_i$ from the values $(\mathbf{x}, \mathbf{u})$ as $F_i(\mathbf{pa}_i, \mathbf{u}_i)$, where[5], $\mathbf{Pa}_i \subseteq \mathbf{X}$ and $\mathbf{U}_i \subseteq \mathbf{U}$. $\mathbf{Pa}_i$ are all endogenous variables that directly influence $X_i$ and $\mathbf{pa}_i$ are their values[6]. $P$ specifies a probability distribution of all exogenous variables $\mathbf{U}$. Without loss of generality, we assume that the domains of exogenous variables are also discrete and finite. The functions $F_i$ are deterministic, i.e., the value of every endogenous variable is uniquely determined given the values of the exogenous variables. Since the exogenous variables follow a probability distribution, this implies a probability distribution over the endogenous variables.

For any basic intervention formula $[X_i{=}x_i]$ (which is our notation for Pearl's do-operator $do(X_i{=}x_i)$), we denote by $\mathcal{F}_{X_i=x_i}$ the functions obtained from $\mathcal{F}$ by replacing $F_i$ with the constant function $F_i(\mathbf{pa}_i, \mathbf{u}_i) := x_i$. We generalize this definition for any intervention $\alpha$ in a natural way and denote as $\mathcal{F}_\alpha$ the resulting functions. For any $\varphi \in \mathcal{E}_{prop}$, we write $\mathcal{F}, \mathbf{u} \models \varphi$ if $\varphi$ is satisfied for the values of $\mathbf{X}$ calculated from the values $\mathbf{u}$. For any intervention $\alpha$, we write $\mathcal{F}, \mathbf{u} \models [\alpha]\varphi$ if $\mathcal{F}_\alpha, \mathbf{u} \models \varphi$. And for all $\psi, \psi_1, \psi_2 \in \mathcal{E}_{counterfact}$, we write $(i)$ $\mathcal{F}, \mathbf{u} \models \neg\psi$ if $\mathcal{F}, \mathbf{u} \not\models \psi$ and $(ii)$ $\mathcal{F}, \mathbf{u} \models \psi_1 \wedge \psi_2$ if $\mathcal{F}, \mathbf{u} \models \psi_1$ and $\mathcal{F}, \mathbf{u} \models \psi_2$. Finally, for $\psi \in \mathcal{E}_{counterfact}$, let $S_\mathfrak{M}(\psi) = \{\mathbf{u} \mid \mathcal{F}, \mathbf{u} \models \psi\}$ be the set of values of $\mathbf{U}$ satisfying $\psi$. For some expression $\mathbf{e}$, we define the value $[\![\mathbf{e}]\!]_\mathfrak{M}$ of the expression $\mathbf{e}$ given a model $\mathfrak{M}$, recursively in a natural way, starting with basic terms as follows $[\![\mathbb{P}(\psi)]\!]_\mathfrak{M} = \sum_{\mathbf{u} \in S_\mathfrak{M}(\psi)} P(\mathbf{u})$ and, for $\delta \in \mathcal{E}_{prop}$, $[\![\mathbb{P}(\psi|\delta)]\!]_\mathfrak{M} = [\![\mathbb{P}(\psi \wedge \delta)]\!]_\mathfrak{M}/[\![\mathbb{P}(\delta)]\!]_\mathfrak{M}$, assuming that the expression is undefined if $[\![\mathbb{P}(\delta)]\!]_\mathfrak{M} = 0$. We will sometimes write $P_\mathfrak{M}(\psi)$ instead of $[\![\mathbb{P}(\psi)]\!]_\mathfrak{M}$, for short. For two expressions $\mathbf{e}_1$ and $\mathbf{e}_2$, we define $\mathfrak{M} \models \mathbf{e}_1 \le \mathbf{e}_2$, iff, $[\![\mathbf{e}_1]\!]_\mathfrak{M} \le [\![\mathbf{e}_2]\!]_\mathfrak{M}$. The semantics for negation and conjunction are defined in the usual way, giving the semantics for $\mathfrak{M} \models \varphi$ for any formula $\varphi$ in $\mathcal{L}_3^*$.

## 2.2 SATISFIABILITY FOR PCH LANGUAGES AND RELEVANT COMPLEXITY CLASSES

The (decision) satisfiability problems for languages of the PCH, denoted by $\text{SAT}_{\mathcal{L}_i}^*$, with $i = 1, 2, 3$ and $* \in Lab$, take as input a formula $\varphi$ in $\mathcal{L}_i^*$ and ask whether there exists a model $\mathfrak{M}$ such that $\mathfrak{M} \models \varphi$. Analogously, the validity problems for $\mathcal{L}_i^*$ consist in deciding whether, for a given $\varphi$, $\mathfrak{M} \models \varphi$ holds for all models $\mathfrak{M}$. From the definitions, it is obvious that variants of the problems for the level $i$ are at least as hard as their counterparts at a lower level.

We note, that the satisfiability problem (and its complement, the validity problem) does not assume anything about SCMs, including their structure. However, our languages allow queries of the form $\psi \Rightarrow \varphi$, which enable us to verify satisfiability, resp., the validity, for the formula $\varphi$ in SCMs which satisfy properties expressed by the formula $\psi$, whereby, e.g., $\psi$ can encode a graph structure of the model. Thus, the formalism used in our work allows for the formulation of a wide range of queries.

To measure the computational complexity of $\text{SAT}_{\mathcal{L}_i}^*$, a central role play the following, well-known Boolean complexity classes NP, PSPACE, NEXP, and EXPSPACE (for formal definitions see, e.g., Arora & Barak (2009)). Recent research has shown that the precise complexity of several natural satisfiability problems can be expressed in terms of the classes over the real numbers $\exists\mathbb{R}$ and $\text{succ-}\exists\mathbb{R}$. For a comprehensive compendium on $\exists\mathbb{R}$, see Schaefer et al. (2024). Recall, that the existential theory of the reals (ETR) is the set of true sentences of the form

$$\exists x_1 \ldots \exists x_n \varphi(x_1, \ldots, x_n), \tag{2}$$

---

[5]We consider recursive models, that is, we assume the endogenous variables are ordered such that variable $X_i$ (i.e. function $F_i$) is not affected by any $X_j$ with $j > i$.

[6]SCMs are often represented as graphs, in which case the variables $\mathbf{Pa}_i$ can be represented as the parents of variable $X_i$. However, the definition using functions does not refer to any graphs.

where $\varphi$ is a quantifier-free Boolean formula over the basis $\{\vee, \wedge, \neg\}$ and a signature consisting of the constants 0 and 1, the functional symbols $+$ and $\cdot$, and the relational symbols $<$, $\leq$, and $=$. The sentence is interpreted over the real numbers in the standard way. The theory forms its own complexity class $\exists\mathbb{R}$ which is defined as the closure of ETR under polynomial time many-one reductions (Grigoriev & Vorobjov, 1988; Canny, 1988; Schaefer, 2009; Schaefer et al., 2024). The significance of this class lies in the exceptional expressiveness of the ETR, enabling the representation of numerous natural problems across computational geometry (Abrahamsen et al., 2018; McDiarmid & Müller, 2013; Cardinal, 2015), Machine Learning and Artificial Intelligence (Abrahamsen et al., 2021; Mossé et al., 2022; van der Zander et al., 2023; Dörfler et al., 2024), game theory (Bilò & Mavronicolas, 2017; Garg et al., 2018), and various other domains.

A succinct variant of ETR, denoted as succ-ETR, and the corresponding class succ-$\exists\mathbb{R}$, have been introduced by van der Zander et al. (2023). succ-ETR is the set of all Boolean circuits $C$ that encode a true sentence as in (2) as follows. Assume that $C$ computes a function $\{0,1\}^N \to \{0,1\}^M$. Then $\{0,1\}^N$ represents the node set of the tree underlying $\varphi$ and $C(i)$ is an encoding of the description of node $i$, consisting of the label of $i$, its parent, and its two children. The variables in $\varphi$ are $x_1, \ldots, x_{2^N}$. As in the case of $\exists\mathbb{R}$, to succ-$\exists\mathbb{R}$ belong all languages which are polynomial time many-one reducible to succ-ETR.

For two computational problems $A, B$, we will write $A \leq_P B$ if $A$ can be reduced to $B$ in polynomial time, which means $A$ is not harder to solve than $B$. A problem $A$ is complete for a complexity class $\mathcal{C}$, if $A \in \mathcal{C}$ and, for every other problem $B \in \mathcal{C}$, it holds $B \leq_P A$. By co-$\mathcal{C}$, we denote the class of all problems $A$ such that their complements $\overline{A}$ belong to $\mathcal{C}$.

## 3 THE INCREASING COMPLEXITY OF SATISFIABILITY IN THE PCH FOR LINEAR LANGUAGES WITH MARGINALIZATION

The expressive power of the languages $\mathcal{L}_i^*$, with $* \in \{base, lin, poly\}$ and the layers $i = 1, 2, 3$ has been the subject of intensive research. It is well known, see e.g. (Pearl, 2009; Bareinboim et al., 2022; Mossé et al., 2022; Suppes & Zanotti, 1981), that they form strict hierarchies along two dimensions: First, on each layer $i$, the languages $\mathcal{L}_i^{base}, \mathcal{L}_i^{lin}$, and $\mathcal{L}_i^{poly}$ have increasing expressiveness; Second, for every $* \in \{base, lin, poly\}$

$$\mathcal{L}_1^* \subsetneq \mathcal{L}_2^* \subsetneq \mathcal{L}_3^* \tag{3}$$

where the proper inclusion means that the language $\mathcal{L}_i^*$ is less expressive than $\mathcal{L}_{i+1}^*$. Note that since adding marginalization does not change the expressiveness of the language $\mathcal{L}_i^*$, the strict inclusions in (3) hold also for $* \in \{base\langle\Sigma\rangle, lin\langle\Sigma\rangle, poly\langle\Sigma\rangle\}$.

To prove such a proper inclusion, it suffices to show two SCMs that are indistinguishable in the less expressive language, but that can be distinguished by some formula in the more expressive language. E.g., the SCMs: $\mathfrak{M} = (\mathcal{F}, P, \mathbf{U}, \mathbf{X})$ and $\mathfrak{M}' = (\mathcal{F}', P, \mathbf{U}, \mathbf{X})$, with binary variables $\mathbf{U} = \{U_1, U_2\}, \mathbf{X} = \{X_1, X_2\}$, probabilities $P(U_i = 0) = 1/2$, and mechanism $\mathcal{F}$: $X_i := U_i$, resp. $\mathcal{F}'$: $X_1 := U_1 U_2 + (1-U_1)(1-U_2)$, $X_2 := U_1 + X_1(1-U_1)U_2$, have the same distributions $P_{\mathfrak{M}}(X_1, X_2) = P_{\mathfrak{M}'}(X_1, X_2)$. Thus, $\mathfrak{M}$ and $\mathfrak{M}'$ are indistinguishable in *any* language of the probabilistic layer. On the other hand, after the intervention $X_1 = 1$, we get $P_{\mathfrak{M}}([X_1{=}1]X_2{=}1) = 1/2$ and $P_{\mathfrak{M}'}([X_1{=}1]X_2{=}1) = 3/4$. Then, e.g., for the $\mathcal{L}_2^{base}$ formula $\varphi$ : $\mathbb{P}([X_1{=}1]X_2{=}1) = \mathbb{P}([X_1{=}1]X_2{=}0)$, we have $\mathfrak{M} \models \varphi$, but $\mathfrak{M}' \not\models \varphi$ which means that $\varphi$ distinguishes $\mathfrak{M}$ from $\mathfrak{M}'$.

This section focuses on the basic and linear languages across the PCH (the case of polynomial languages will be discussed in Sec. 4 separately). As mentioned in the introduction, a comparison of these languages from the perspective of computational complexity reveals surprisingly different properties than the ones described above. For basic and linear languages disallowing marginalization, the satisfiability for the counterfactual level remains the same as for the probabilistic and causal levels: problems $\text{SAT}_{\mathcal{L}_i}^{base}$ and $\text{SAT}_{\mathcal{L}_i}^{lin}$ for all $i = 1, 2, 3$ are NP-complete, i.e., as hard as reasoning about propositional logic formulas ((Fagin et al., 1990; Mossé et al., 2022), cf. also Table 1). In this section, we show that the situation changes drastically when marginalization is allowed: satisfiability problems $\text{SAT}_{\mathcal{L}_i}^{base\langle\Sigma\rangle}$ and $\text{SAT}_{\mathcal{L}_i}^{lin\langle\Sigma\rangle}$ become $\text{NP}^{\text{PP}}$-, PSPACE-, resp. NEXP-complete depending on the level $i$. This demonstrates the first strictly increasing complexity of reasoning across the PCH, assuming the widely accepted assumption that the inclusions $\text{NP}^{\text{PP}} \subseteq \text{PSPACE} \subseteq \text{NEXP}$ are proper.

### 3.1 THE PROBABILISTIC (OBSERVATIONAL) LEVEL

Marginalization via the summation operator, combined with the language expressing events $\delta$ on a specific level of the PCH, increases the complexity of reasoning to varying degrees, depending on the level. In the probabilistic case, it jumps from NP- to $\mathsf{NP}^{\mathsf{PP}}$-completeness since the atomic terms $\mathbb{P}(\delta)$ can contain Boolean formulas $\delta$, which, combined with the summation operator, allows to count the number of all satisfying assignments of a Boolean formula by summing over all possible values for the random variables in the formula. Determining this count is the canonical PP-complete problem, so evaluating the equations given a model is PP-hard. From this, together with the need of finding a model, we will conclude in this section that $\mathrm{SAT}_{\mathcal{L}_1}^{base\langle\Sigma\rangle}$ and $\mathrm{SAT}_{\mathcal{L}_1}^{lin\langle\Sigma\rangle}$ are $\mathsf{NP}^{\mathsf{PP}}$-complete.

We start with a technical but useful fact that a sum in the probabilistic language can be partitioned into a sum over probabilities and a sum over purely logical terms. This generalizes the property shown by Fagin et al. (1990) in Lemma 2.3 for languages without the summation operator.

**Fact 2.** *Let $\delta \in \mathcal{E}_{prop}$ be a propositional formula over variables $X_{i_1}, \ldots, X_{i_l}$. A sum $\sum_{x_{i_1}} \cdots \sum_{x_{i_l}} \mathbb{P}(\delta)$ is equal to $\sum_{\hat{x}_1} \cdots \sum_{\hat{x}_n} p_{\hat{x}_1 \ldots \hat{x}_n} \sum_{x_{i_1}} \cdots \sum_{x_{i_l}} \delta_{\hat{x}_1 \ldots \hat{x}_n}(x_{i_1}, \ldots, x_{i_l})$ where the range of the sums is the entire domain, $p_{\hat{x}_1 \ldots \hat{x}_n}$ is the probability of $\mathbb{P}(X_1 = \hat{x}_1 \wedge \ldots \wedge X_n = \hat{x}_n)$ and $\delta_{\hat{x}_1 \ldots \hat{x}_n}(x_{i_1}, \ldots, x_{i_l})$ a function that returns 1 if the implication $(X_1 = \hat{x}_1 \wedge \ldots \wedge X_n = \hat{x}_n) \rightarrow \delta(x_{i_1}, \ldots, x_{i_l})$ is a tautology and 0 otherwise.*

The canonical satisfiability problem SAT, where instances consist of Boolean formulas in propositional logic, plays a key role in a broad range of research fields, since all problems in NP, including many real-world tasks can be naturally reduced to it. As discussed earlier, $\mathrm{SAT}_{\mathcal{L}_1}^{base}$ and $\mathrm{SAT}_{\mathcal{L}_1}^{lin}$ do not provide greater modeling power than SAT. Enabling the use of summation significantly changes this situation: below we demonstrate the expressiveness of $\mathrm{SAT}_{\mathcal{L}_1}^{base\langle\Sigma\rangle}$, showing that any problem in $\mathsf{NP}^{\mathsf{PP}}$ can be reduced to it in polynomial time.

**Lemma 3.** $\mathrm{SAT}_{\mathcal{L}_1}^{base\langle\Sigma\rangle}$ *is $\mathsf{NP}^{\mathsf{PP}}$-hard.*

*Proof.* The canonical $\mathsf{NP}^{\mathsf{PP}}$-complete problem E-MajSat is deciding the satisfiability of a formula $\psi : \exists x_1 \ldots x_n : \#y_1 \ldots y_n \phi \geq 2^{n-1}$, i.e. deciding whether the Boolean formula $\phi$ has a majority of satisfying assignments to Boolean variables $y_i$ after choosing Boolean variables $x_i$ existentially (Littman et al., 1998). We will reduce this to $\mathrm{SAT}_{\mathcal{L}_1}^{base\langle\Sigma\rangle}$. Let there be $2n$ random variables $X_1, \ldots, X_n, Y_1, \ldots, Y_n$ associated with the Boolean variables. Assume w.l.o.g. that these random variables have domain $\{0, 1\}$. Let $\phi'$ be the formula $\phi$ after replacing Boolean variable $x_i$ with $X_i = 0$ and $y_i$ with $Y_i = y_i$.

Consider the probabilistic inequality $\varphi : \sum_{y_1} \cdots \sum_{y_n} \mathbb{P}(\phi') \geq 2^{n-1}$ whereby $2^{n-1}$ is encoded as $\sum_{x_1} \cdots \sum_{x_{n-1}} \mathbb{P}(\top)$. The left hand side equals

$$\sum_{\hat{x}_1} \cdots \sum_{\hat{x}_n} \sum_{\hat{y}_1} \cdots \sum_{\hat{y}_n} p_{\hat{x}_1 \ldots \hat{x}_n, \hat{y}_1 \ldots \hat{y}_n} \sum_{y_1} \cdots \sum_{y_n} \delta_{\hat{x}_1 \ldots \hat{x}_n, \hat{y}_1 \ldots \hat{y}_n}(y_1, \ldots, y_n)$$

according to Fact 2. Since the last sum ranges over all values of $y_i$, it counts the number of satisfying assignments to $\phi$ given $\hat{x}_i$. Writing this count as $\#_y(\phi(\hat{x}_1 \ldots \hat{x}_n))$, the expression becomes: $\sum_{\hat{x}_1} \cdots \sum_{\hat{x}_n} \sum_{\hat{y}_1} \cdots \sum_{\hat{y}_n} p_{\hat{x}_1 \ldots \hat{x}_n, \hat{y}_1 \ldots \hat{y}_n} \#_y(\phi(\hat{x}_1 \ldots \hat{x}_n))$.

Since $\#_y(\phi(\hat{x}_1 \ldots \hat{x}_n))$ does not depend on $\hat{y}$, we can write the expression as $\sum_{\hat{x}_1} \cdots \sum_{\hat{x}_n} p_{\hat{x}_1 \ldots \hat{x}_n} \#_y(\phi(\hat{x}_1 \ldots \hat{x}_n))$ whereby $p_{\hat{x}_1 \ldots \hat{x}_n} = \sum_{\hat{y}_1} \cdots \sum_{\hat{y}_n} p_{\hat{x}_1 \ldots \hat{x}_n, \hat{y}_1 \ldots \hat{y}_n}$.

If $\psi$ is satisfiable, there is an assignment $\hat{x}_1, \ldots, \hat{x}_n$ with $\#_y(\phi(\hat{x}_1 \ldots \hat{x}_n)) \geq 2^{n-1}$. If we set $p_{\hat{x}_1 \ldots \hat{x}_n} = 1$ and every other probability $p_{\hat{x}'_1 \ldots \hat{x}'_n} = 0$, $\varphi$ is satisfied.

If $\varphi$ is satisfiable, let $x_1^{max}, \ldots, x_n^{max}$ be the assignment that maximizes $\#_y(\phi(x_1^{max} \ldots x_n^{max}))$. Then $2^{n-1} \leq \sum_{\hat{x}_1} \cdots \sum_{\hat{x}_n} p_{\hat{x}_1 \ldots \hat{x}_n} \#_y(\phi(\hat{x}_1 \ldots \hat{x}_n)) \leq \sum_{\hat{x}_1} \cdots \sum_{\hat{x}_n} p_{\hat{x}_1 \ldots \hat{x}_n} \#_y(\phi(x_1^{max} \ldots x_n^{max})) = \left(\sum_{\hat{x}_1} \cdots \sum_{\hat{x}_n} p_{\hat{x}_1 \ldots \hat{x}_n}\right) \#_y(\phi(x_1^{max} \ldots x_n^{max})) = \#_y(\phi(x_1^{max} \ldots x_n^{max}))$. Hence $\psi$ is satisfied for $x_1^{max} \ldots x_n^{max}$. $\square$

**Theorem 4.** *For probabilistic reasoning, the satisfiability problems $\mathrm{SAT}_{\mathcal{L}_1}^{base\langle\Sigma\rangle}$ and $\mathrm{SAT}_{\mathcal{L}_1}^{lin\langle\Sigma\rangle}$, for the basic and linear languages respectively, are $\mathsf{NP}^{\mathsf{PP}}$-complete.*

*Proof idea:* Having Lemma 3, it remains to prove that the problem is in $\mathsf{NP}^{\mathsf{PP}}$. To this aim, we show that any satisfiable instance has a solution of polynomial size: we rewrite the sums $\sum_{x_{i_1}} \cdots \sum_{x_{i_l}} \mathbb{P}(\delta)$ in the expressions according to Fact 2 which allows to encode the instance as a system of $m$ linear equations with unknown coefficients $p_{\hat{x}_1 \ldots \hat{x}_n}$, where $m$ is bounded by the instance size. Such a system has a non-negative solution with at most $m$ entries positive of polynomial size. Then, we guess non-deterministically such solutions and verify its correctness estimating the values $\sum_{x_{i_1}} \ldots \sum_{x_{i_l}} \delta_{\hat{x}_1 \hat{x}_n}(x_{i_1}, .., x_{i_l})$ using the PP oracle. A full proof, as for all further proof ideas, can be found in the appendix. $\qquad\square$

**Remark 5.** *$\mathsf{NP}^{\mathsf{PP}}$ is the class describing the complexity of another, relevant primitive of the probabilistic reasoning which consists in finding the Maximum a Posteriori Hypothesis (MAP). To study its computational complexity, the corresponding decision problem is defined which asks if for a given Bayesian network $\mathcal{B} = (\mathcal{G}, P_\mathcal{B})$, where probability $P_\mathcal{B}$ factorizes according to the structure of network $\mathcal{G}$, a rational number $\tau$, evidence $\mathbf{e}$, and some subset of variables $\mathbf{Q}$, there is an instantiation $\mathbf{q}$ to $\mathbf{Q}$ such that $P_\mathcal{B}(\mathbf{q}, \mathbf{e}) = \sum_\mathbf{y} P_\mathcal{B}(\mathbf{q}, \mathbf{y}, \mathbf{e}) > \tau$. It is well known that the problem is $\mathsf{NP}^{\mathsf{PP}}$-complete (Roth, 1996; Park & Darwiche, 2004).*

Finally, we draw our attention to the impact of negation on the complexity of reasoning for such languages of the probabilistic layer. The hardness of $\mathrm{SAT}_{\mathcal{L}_1}^{lin\langle\Sigma\rangle}$ depends on negations in the Boolean formulas $\delta$ of basic terms $\mathbb{P}(\delta)$, which make it difficult to count all satisfying assignments. Without negations, marginalization just removes variables, e.g. $\sum_x \mathbb{P}(X{=}x \wedge Y{=}y)$ becomes $\mathbb{P}(Y{=}y)$. This observation leads to the following:

**Proposition 6.** $\mathrm{SAT}_{\mathcal{L}_1}^{base\langle\Sigma\rangle}$ *and* $\mathrm{SAT}_{\mathcal{L}_1}^{lin\langle\Sigma\rangle}$ *are* $\mathsf{NP}$-*complete if the primitives in $\mathcal{E}_{post\text{-}int}$ are restricted to not contain negations.*

## 3.2 The Interventional (Causal) Level

In causal formulas, one can perform interventions to set the value of a variable, which can recursively affect the value of all endogenous variables that depend on the intervened variable. This naturally corresponds to the choice of a variable value by an existential or universal quantifier in a Boolean formula, since in a Boolean formula with multiple quantifiers, the value chosen by each quantifier can depend on the values chosen by earlier quantifiers. Thus, an interventional equation can encode a quantified Boolean formula. This makes $\mathrm{SAT}_{\mathcal{L}_2}^{lin\langle\Sigma\rangle}$ PSPACE-hard. As we will show below, it is even PSPACE-complete.

**Lemma 7.** $\mathrm{SAT}_{\mathcal{L}_2}^{base\langle\Sigma\rangle}$ *is* PSPACE *hard.*

*Proof.* We reduce from the canonical PSPACE-complete problem QBF. Let $Q_1 x_1 Q_2 x_2 \cdots Q_n x_n \psi$ be a quantified Boolean formula with arbitrary quantifiers $Q_1, \ldots, Q_n \in \{\exists, \forall\}$. We introduce Boolean random variables $\mathbf{X} = \{X_1, \ldots, X_n\}$ to represent the values of the variables and denote by $\mathbf{Y} = \{Y_1, \ldots Y_k\} \subseteq \mathbf{X}$ the universally quantified variables. In $\mathcal{L}_2$ one can enforce an ordering $X_1 \prec X_2 \prec \ldots \prec X_n$ of variables, i.e. variable $X_i$ can only depend on variables $X_j$ with $j < i$. The proof of this property, formulated in Lemma 13, can be found in the appendix. Our only further constraint is

$$\sum_\mathbf{y} \mathbb{P}([\mathbf{y}]\psi') = 2^k \qquad (4)$$

where $\psi'$ is obtained from $\psi$ by replacing positive literals $x_i$ by $X_i = 1$ and negative literals $\overline{x_i}$ by $X_i = 0$.

Suppose the constructed $\mathrm{SAT}_{\mathcal{L}_2}^{lin\langle\Sigma\rangle}$ formula is satisfied by a model $\mathfrak{M}$. We show that $Q_1 x_1 Q_2 x_2 \cdots Q_n x_n \psi$ is satisfiable. Each probability implicitly sums over all possible values $\mathbf{u}$ of the exogenous variables. Fix one such $\mathbf{u}$ with positive probability. Combined with $X_1 \prec \ldots \prec X_n$ this implies all random variables now deterministically depend only on any of the previous variables. Equation (4) enforces $\mathbb{P}([\mathbf{y}]\psi') = 1$ for every choice of $\mathbf{y}$ and thus simulates the $\mathbf{Y}$ being universally quantified. As the existential variables $x_i$, we then choose the value $x_i$ of $X_i$ which can only depend on $X_j$ with $j < i$. The formula $\psi'$ and thus $\psi$ is then satisfied due to $\mathbb{P}([\mathbf{y}]\psi') = 1$.

On the other hand, suppose $Q_1 x_1 Q_2 x_2 \cdots Q_n x_n \psi$ is satisfiable, We create a deterministic model $\mathfrak{M}$ as follows: The value of existentially quantified variables $X_i$ is then computed by the function

$F_i(x_1, \ldots, x_{i-1})$ defined as the existentially chosen value when the previous variables are set to $x_1, \ldots, x_{i-1}$. The values of the universally quantified variables do not matter since we intervene on them before every occurrence. This satisfies the required order of variables and, since $\psi$ is satisfied, we have $\mathbb{P}([\mathbf{y}]\psi') = 1$, for every choice of the universally quantified variables $\mathbf{y}$, thus satisfying equation (4). $\qquad\square$

**Theorem 8.** *For causal reasoning, the satisfiability problems* $\text{SAT}_{\mathcal{L}_2}^{base\langle\Sigma\rangle}$ *and* $\text{SAT}_{\mathcal{L}_2}^{lin\langle\Sigma\rangle}$, *for the basic and linear languages respectively, are* PSPACE-*complete.*

*Proof idea:* After Lemma 7, we only have to show that $\text{SAT}_{\mathcal{L}_2}^{lin\langle\Sigma\rangle}$ is in PSPACE. For the proof details, see the appendix. The basic idea is to evaluate interventions one at a time without storing the entire model because each primitive can only contain one intervention. For each sum, we only need to store the total probability of all its primitives, and increment this probability, if a new intervention satisfies some of the primitives.

As with $\text{SAT}_{\mathcal{L}_1}^{lin\langle\Sigma\rangle}$, there can only be polynomially many exogenous variable assignments $\mathbf{u}$ with non-zero probability $p_{\mathbf{u}}$, which are independent of each other and can be guessed non-deterministically. We can also guess a causal order of the endogenous variables, such that variables can only depend on the variables preceding them in the causal order. This causal order allows one to guess the variables affected by any intervention in a sound way.

We enumerate all exogenous variable assignments $\mathbf{u}$, each having probability $p_{\mathbf{u}}$. Recursively, we can enumerate all possible interventions $\boldsymbol{\alpha}$, which yield the values $\mathbf{x}$ of the endogenous variables, which—given the exogenous variables—also have probability $p_{\mathbf{u}}$. Then we count for each sum how many of its primitives are satisfied by $\boldsymbol{\alpha}$ and $\mathbf{x}$, and increment the accumulated value of the sum by $p_{\mathbf{u}}$ for each.

After this enumeration, we know the numeric value of every sum, and can verify the resulting equation system. $\qquad\square$

### 3.3 THE COUNTERFACTUAL LEVEL

On the counterfactual level, one can perform multiple interventions, and thus compare different functions of the model to each other. Hence, the formulas can only be evaluated if the entire, exponential-sized model is known. Thus deciding the satisfiability requires exponential time and non-determinism to find the model, making $\text{SAT}_{\mathcal{L}_3}^{lin\langle\Sigma\rangle}$ NEXP-complete.

**Theorem 9.** *For counterfactual reasoning, the satisfiability problems* $\text{SAT}_{\mathcal{L}_3}^{base\langle\Sigma\rangle}$ *and* $\text{SAT}_{\mathcal{L}_3}^{lin\langle\Sigma\rangle}$, *for the basic and linear languages respectively, are* NEXP-*complete.*

*Proof idea:* This NEXP-hardness follows from a reduction from the NEXP-complete problem of checking satisfiability of a Schönfinkel-Bernays sentence to the satisfiability of $\text{SAT}_{\mathcal{L}_3}^{base\langle\Sigma\rangle}$. These are first-order logic formulas of the form $\exists\mathbf{x}\forall\mathbf{y}\psi$ where $\psi$ cannot contain any quantifiers or functions. The $\mathbf{x}$ and $\mathbf{y}$ are encoded as Boolean random variables, where the existentially quantified $\mathbf{x}$ are encoded into the existence of a satisfying SCM, while the universally quantified $\mathbf{y}$ are encoded by marginalization, i.e. a condition $\sum_{\mathbf{y}} \mathbb{P}([\mathbf{y}]\psi') = 2^n$ for some formula $\psi'$ derived from $\psi$ and where $n$ denotes the number of variables $\mathbf{y}$. The counterfactuals now allow us to ensure that the random variables $R_i$ representing the relations within $\psi$ deterministically depend on their respective inputs by comparing whether $R_i$ changes between an intervention on all variables (except $R_i$) versus an intervention on only its dependencies. Marginalization finally allows us to combine these exponentially many checks for all possible values of all variables into a single equation.

The containment in NEXP follows from expanding the formulas with sums to exponentially larger formulas without sums. $\qquad\square$

## 4 THE COMPLEXITY OF SATISFIABILITY FOR POLYNOMIAL LANGUAGES WITH MARGINALIZATION

Van der Zander et al. (2023) prove that $\text{SAT}_{\mathcal{L}_1}^{poly\langle\Sigma\rangle}$ and $\text{SAT}_{\mathcal{L}_2}^{poly\langle\Sigma\rangle}$ are complete for succ-$\exists\mathbb{R}$ whenever the basic terms are allowed to also contain conditional probabilities. They, however, leave open the

exact complexity status of $\text{SAT}_{\mathcal{L}_3}^{poly\langle\Sigma\rangle}$. Here we show that the problem is in succ-$\exists\mathbb{R}$, with and without conditional probabilities, which proves its succ-$\exists\mathbb{R}$-completeness.

**Theorem 10.** $\text{SAT}_{\mathcal{L}_3}^{poly\langle\Sigma\rangle} \in$ succ-$\exists\mathbb{R}$. *This also holds true if we allow the basic terms to contain conditional probabilities.*

*Proof idea:* Bläser et al. (2024) introduced the $\text{NEXP}_{real}$ machine model, where succ-$\exists\mathbb{R}$ is precisely the set of all languages decidable by exponential-time non-deterministic real RAMs. Combined with the algorithm proving $\text{SAT}_{\mathcal{L}_3}^{poly} \in \exists\mathbb{R}$ (however without subtractions or conditional probabilities) from Mossé et al. (2022) and the classification $\exists\mathbb{R} = \text{NP}_{real}$ from Erickson et al. (2022), i.e. $\exists\mathbb{R}$ being the set of all languages decidable by polynomial-time non-deterministic real RAMs, we can expand the unary sums explicitly and then run the non-deterministic real RAM algorithm for the resulting $\text{SAT}_{\mathcal{L}_3}^{poly}$ instance. Special care has to be taken in dealing with subtractions or conditional probabilities, here we use a trick by Tseitin, the details of which can be found in Lemma 15 in the appendix. $\quad\square$

We note that Ibeling et al. (2024, Theorem 3) independently obtained a variant of the above result, also using the machine characterization of succ-$\exists\mathbb{R}$ given by Bläser et al. (2024).

**Remark 11.** *It can be shown that the hardness proofs for* $\text{SAT}_{\mathcal{L}_1}^{poly\langle\Sigma\rangle}$, $\text{SAT}_{\mathcal{L}_2}^{poly\langle\Sigma\rangle}$, *and* $\text{SAT}_{\mathcal{L}_3}^{poly\langle\Sigma\rangle}$ *do not need conditional probabilities in the basic terms.*

## 5 DISCUSSION

This work studies the computational complexities of satisfiability problems for languages at all levels of the PCH. Our new completeness results nicely extend and complement the previous achievements by Fagin et al. (1990), Mossé et al. (2022), van der Zander et al. (2023), and Bläser et al. (2024). The main focus of our research was on languages allowing the use of marginalization which is expressed in the languages by a summation operator $\Sigma$ over the domain of the random variables. This captures the standard notation commonly used in probabilistic and causal inference.

A very interesting feature of the satisfiability problems for the full, polynomial languages is the following property. For both variants, with and without summation operators, while the expressive powers of the corresponding languages differ, the complexities of the corresponding satisfiability problems at all three levels of the PCH are the same. Interestingly, the same holds for linear languages *without* marginalization, too (cf. Table 1.). We find that the situation changes drastically in the case of linear languages *allowing* the summation operator $\Sigma$. One of our main results characterizes the complexities of $\text{SAT}_{\mathcal{L}_1}^{lin\langle\Sigma\rangle}$, $\text{SAT}_{\mathcal{L}_2}^{lin\langle\Sigma\rangle}$, and $\text{SAT}_{\mathcal{L}_3}^{lin\langle\Sigma\rangle}$ problems as $\text{NP}^{\text{PP}}$, PSPACE, and NEXP-complete, respectively. The analogous completeness results hold for $\text{SAT}_{\mathcal{L}_i}^{base\langle\Sigma\rangle}$.

Another interesting feature is that the completeness results for linear languages are expressed in terms of standard Boolean classes while the completeness of satisfiability for languages involving polynomials over the probabilities requires classes over the reals.

As mentioned in Section 2.2, the formulas can also encode graph structures. One might ask whether the computational complexity changes if the graph is not encoded in the formulas but is given directly as a graph in the input, which is a common setting in causal reasoning. Also to show the complexity upper-bound on the probabilistic and interventional linear languages, we used a fact that it is sufficient to only consider models of polynomial size for these languages (cf. Footnote 8 in the appendix). Here one might wonder what is the complexity of restricting oneself to polynomially-sized models for the other languages? We answer both questions in a follow-up paper (Bläser et al., 2025).

ACKNOWLEDGMENTS

This work was supported by the Deutsche Forschungsgemeinschaft (DFG) grant 471183316 (ZA 1244/1-1). We thank Till Tantau for the key idea that made the proof of the $\text{NP}^{\text{PP}}$-hardness of $\text{SAT}_{\mathcal{L}_1}^{lin\langle\Sigma\rangle}$ possible.

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

# A  TECHNICAL DETAILS AND PROOFS

In this section we complete proofs of our main results (announced in Table 1 in the Introduction): Theorem 4, 8, 9, and Theorem 10, which were partially presented or outlined in the main part of the paper. More precisely, Theorem 4 will follow from Lemma 3 and 12, Theorem 8 from Lemma 7 and 14, Theorem 9 will be proved in Sec. A.3, and the proof of Theorem 10 will be presented in Sec. A.4. We also show the proof of Fact 2 and of Proposition 6.

## A.1  PROOFS OF SECTION 3.1

We first prove Fact 2. Theorem 4 will follow from Lemma 3 and Lemma 12 presented below.

*Proof of Fact 2.* $\sum_{x_{i_1}} \ldots \sum_{x_{i_l}} \mathbb{P}(\delta)$ is equivalent to

$$
\sum_{x_{i_1}} \cdots \sum_{x_{i_l}} \sum_{\hat{x}_1} \cdots \sum_{\hat{x}_n} \mathbb{P}((X_1 = \hat{x}_1 \wedge \ldots \wedge X_n = \hat{x}_n) \wedge \delta)
$$

$$
= \sum_{\hat{x}_1} \cdots \sum_{\hat{x}_n} \sum_{x_{i_1}} \cdots \sum_{x_{i_l}} \mathbb{P}((X_1 = \hat{x}_1 \wedge \ldots \wedge X_n = \hat{x}_n) \wedge \delta)
$$

$$
= \sum_{\hat{x}_1} \cdots \sum_{\hat{x}_n} \sum_{x_{i_1}} \cdots \sum_{x_{i_l}} p_{\hat{x}_1 \ldots \hat{x}_n} \delta_{\hat{x}_1 \ldots \hat{x}_n}(x_{i_1}, \ldots, x_{i_l})
$$

$$
= \sum_{\hat{x}_1} \cdots \sum_{\hat{x}_n} p_{\hat{x}_1 \ldots \hat{x}_n} \sum_{x_{i_1}} \cdots \sum_{x_{i_l}} \delta_{\hat{x}_1 \ldots \hat{x}_n}(x_{i_1}, \ldots, x_{i_l}), \tag{5}
$$

which completes the proof. □

Since the second sums in (5) only depend on $\hat{x}_1 \ldots \hat{x}_n$ and not on $p_{\hat{x}_1 \ldots \hat{x}_n}$, they can be calculated without knowing the probability distribution. Due to the dependency on $\hat{x}_1 \ldots \hat{x}_n$ the expression cannot be simplified further in general. However, if event $\delta$ in $\mathbb{P}(\delta)$ contains no constant events like $X_{i_j} = 0$ but only events $X_{i_j} = x_{i_j}$ depending on the summation variables $x_{i_j}$, the sum always includes one iteration where the event occurs and $c - 1$ iterations where it does not. Thus the sum $\sum_{x_{i_1}} \ldots \sum_{x_{i_l}} \delta_{\hat{x}_1 \ldots \hat{x}_n}(x_{i_1}, \ldots, x_{i_l})$ is constant and effectively counts the number of satisfying assignments to the formula $\delta$. Since the sum $\sum_{\hat{x}_1} \ldots \sum_{\hat{x}_n} p_{\hat{x}_1 \ldots \hat{x}_n}$ is always 1, the original sum $\sum_{x_{i_1}} \ldots \sum_{x_{i_l}} \mathbb{P}(\delta)$ also counts the number of satisfying assignments.

**Lemma 12.** $\text{SAT}_{\mathcal{L}_1}^{lin\langle\Sigma\rangle}$ *is in* $\text{NP}^{\text{PP}}$.

*Proof.* First we need to show that satisfiable instances have solutions of polynomial size.

We write each (sum of a) primitive in the arithmetic expressions as

$$
\sum_{\hat{x}_1} \cdots \sum_{\hat{x}_n} p_{\hat{x}_1 \ldots \hat{x}_n} \sum_{x_{i_1}} \cdots \sum_{x_{i_l}} \delta_{\hat{x}_1 \ldots \hat{x}_n}(x_{i_1}, \ldots, x_{i_l})
$$

according to Fact 2.

The value of all $p_{\hat{x}_1 \ldots \hat{x}_n}$ can be encoded as a vector $\vec{p} \in \mathbb{R}^{c^n}$. For each $\hat{x}_1, \ldots, \hat{x}_n$, the sum $\sum_{x_{i_1}} \ldots \sum_{x_{i_l}} \delta_{\hat{x}_1 \ldots \hat{x}_n}(x_{i_1}, \ldots, x_{i_l})$ is a constant integer $\leq c^l$. Suppose there are $m$ such sums or primitives in the instance, whose values can be encoded as a matrix $A \in \mathbb{R}^{m \times c^n}$.

Then the value of every sum in the instance is given by the vector $A\vec{p} \in \mathbb{R}^m$.

There exists a non-negative vector $\vec{q} \in \mathbb{Q}^{c^n}$ containing at most $m$ non-zero entries with $A\vec{q} = A\vec{p}$ (Lemma 2.5 in Fagin et al. (1990)). By including a constraint $\sum_{\hat{x}_1} \ldots \sum_{\hat{x}_n} p_{\hat{x}_1 \ldots \hat{x}_n} = 1$ when constructing the matrix $A$, we can ensure that all values in $\vec{q}$ are valid probabilities. $\vec{q}$ is a polynomial sized solution and can be guessed non-deterministically.

For this solution, the sum $\sum_{\hat{x}_1} \ldots \sum_{\hat{x}_n} p_{\hat{x}_1 \ldots \hat{x}_n}$ can then be evaluated (over all non-zero entries $p_{\hat{x}_1 \ldots \hat{x}_n}$) in polynomial time. Each sum $\sum_{x_{i_1}} \ldots \sum_{x_{i_l}} \delta_{\hat{x}_1 \ldots \hat{x}_n}(x_{i_1}, \ldots, x_{i_l})$ can be evaluated using

the PP oracle because a PP oracle is equivalent to a #P oracle which can count the number of satisfying assignments of $\delta$. $\square$

Finally, we give the proof of Proposition 6.

*Proof of Proposition 6.* In the proof of Lemma 12, we require a PP-oracle to evaluate the sum $\sum_{x_{i_1}} \cdots \sum_{x_{i_l}} \delta_{\hat{x}_1 \dots \hat{x}_n}(x_{i_1}, \dots, x_{i_l})$, which counts the number of assignments $x_{i_1}, \dots, x_{i_l}$ that make $(X_1 = \hat{x}_1 \wedge \dots \wedge X_n = \hat{x}_n) \rightarrow \delta(x_{i_1}, \dots, x_{i_l})$ a tautology for given $\hat{x}_i$.

Without negations, there are also no disjunctions, so $\delta$ consists of a conjunction of terms that compare some variable to some constant, $X_i = j$, or to some variable $X_i = x_{i_j}$ used in the summation. For any constant $X_i = j$, we check whether $\hat{x}_i = j$. If that is false, the number of satisfying assignments to $\delta$ is zero. From any condition $X_i = x_{i_j}$, we learn that $x_{i_j}$ has to be equal to $\hat{x}_i$. If there is any contradiction, i.e., $X_i = x_{i_j} \wedge X_k = x_{i_j}$ and $\hat{x}_i \neq \hat{x}_k$, the number of assignments is also zero. This determines the value of each $x_{i_j}$ occurring in $\delta$ and makes $\delta$ a tautology.

It leaves the value of $x_{i_j}$ not occurring in $\delta$ undefined, but those do not affect $\delta$, and can be chosen arbitrarily, so the number of assignments is just multiplied by the size of the domain $c$ for each not-occurring variable.

This can be evaluated in polynomial time, so the complexity is reduced to $\mathsf{NP}^P = \mathsf{NP}$.

The problems remain NP-hard since a Boolean formula can be encoded in the language $\mathcal{L}_1^{base}$ of combinations of Boolean inequalities. Each Boolean variable $x$ is replaced by $P(X) > 0$ for a corresponding random variable $X$. For example, a 3-SAT instance like $(x_{i_{1,1}} \vee \neg x_{i_{1,2}} \vee x_{i_{1,3}}) \wedge (x_{i_{2,1}} \vee \neg x_{i_{2,2}} \vee \neg x_{i_{2,3}}) \wedge \dots$ can be encoded as $(P(X_{i_{1,1}}) > 0 \vee \neg P(X_{i_{1,2}}) > 0 \vee P(X_{i_{1,3}}) > 0) \wedge (P(X_{i_{2,1}}) > 0 \vee \neg P(X_{i_{2,2}}) > 0 \vee \neg P(X_{i_{2,3}}) > 0) \wedge \dots$, which clearly has the same satisfiability. $\square$

## A.2 PROOFS OF SECTION 3.2

We first prove the property stated in lemma below.

**Lemma 13.** *In $\mathcal{L}_2$ one can encode a causal ordering.*

*Proof.* Given (in-)equalities in $\mathcal{L}_2$, we add a new variable $C$ and add to each primitive the intervention $[C = 0]$. This does not change the satisfiability of (in-)equalities.

Given a causal order $V_{i_1} \prec V_{i_2} \prec \dots$, we add $c$ equations for each variable $V_{i_j}, j > 1$:

$$\mathbb{P}([C{=}1, V_{i_{j-1}}{=}k]V_{i_j}{=}k) = 1 \text{ for } k = 1, \dots, c.$$

The equations ensure, that if one variable is changed, and $C = 1$ is set, the next variable in the causal ordering has the same value, thus fixing an order from the first to the last variable. $\square$

Theorem 8 follows from Lemma 7 and from:

**Lemma 14.** $\mathrm{SAT}_{\mathcal{L}_2}^{lin\langle\Sigma\rangle}$ *is in* PSPACE.

*Proof.* We need to show that Algorithm 1 is correct and in PSPACE. The basic idea of the algorithm is that rather than guessing a model and evaluating each sum with its interventions[7], we enumerate all possible interventions (and resulting values) and increment each sum that includes the intervention. Thereby, rather than storing the functions and interventions, we only need to store and update the value of the sums.

By definition, each sum $\sum_{\mathbf{y}_i} \mathbb{P}([\alpha_i]\delta_i)$ in the input can be written as $\sum_{\mathbf{u}} p_{\mathbf{u}} \sum_{\mathbf{y}_i : \mathcal{F}, \mathbf{u} \models [\alpha_i]\delta_i} 1$, where the second sum does not depend on $p_{\mathbf{u}}$. As in Lemma 12, one can write the probabilities as a single vector $\vec{p_{\mathbf{u}}}$, each sum $\sum_{\mathbf{y}_i : \mathcal{F}, \mathbf{u} \models [\alpha_i]\delta_i} 1$ as row in a matrix $A$, such that an entry of the row is 1 if $\mathcal{F}, \mathbf{u} \models [\alpha_i]\delta_i$ holds and 0 otherwise. Then one obtains the value of all sums as product $A\vec{p_{\mathbf{u}}}$. A

---

[7]For example, a sum like $\sum_x \mathbb{P}([X{=}x]Y{=}y)$ performs multiple interventions on $X$, which is difficult to evaluate. Sums containing only a single intervention could be evaluated trivially.

**Input:** $\text{SAT}_{\mathcal{L}_2}^{lin\langle\Sigma\rangle}$ instance
**Output:** Is the instance satisfiable?

1 Guess small model probabilities $p_{\mathbf{u}}$;
2 Guess a causal order $X_1, \ldots, X_n$;
3 Rewrite each sum $\sum_{\mathbf{y}_i} \mathbb{P}([\alpha_i]\delta_i)$ in the input as $\sum_{\mathbf{u}} p_{\mathbf{u}} \sum_{\mathbf{y}_i:\mathcal{F}, \mathbf{u}\models[\alpha_i]\delta_i} 1$;
4 Initialize a counter $c_i$ to zero for each such sum
5 **for** $p_{\mathbf{u}} > 0$ **do**
6 $\quad$ Guess values $x_1, .., x_n$;
7 $\quad$ Simulate-Interventions$(1, \{\}, x_1, ..., x_n)$
8 **end**
9 Replace the sums by $c_i$ and verify whether the (in-)equalities are satisfied;

10 **Function** *Simulate-Interventions*$(i, \boldsymbol{\alpha}, x_1, \ldots, x_n)$
$\quad$ **Input:** Current variable $X_i$; set of interventions $\alpha$; values $x_1, \ldots, x_n$
11 $\quad$ **if** $i > n$ **then**
12 $\quad\quad$ **for** *each sum counter $c_j$* **do**
13 $\quad\quad\quad$ **for** *all possible values $\mathbf{y}_j$ of the sum* **do**
14 $\quad\quad\quad\quad$ **if** $\alpha_j$ *(after inserting $\mathbf{y}_i$) is $[\{X_i = x_i \text{ for } i \in \boldsymbol{\alpha}\}]$ and $x_1, .., x_n$ satisfy $\delta_j$ (after inserting $\mathbf{y}_i$)* **then**
15 $\quad\quad\quad\quad\quad$ increment $c_j$ by $p_{\mathbf{u}}$
16 $\quad\quad\quad\quad$ **end**
17 $\quad\quad\quad$ **end**
18 $\quad\quad$ **end**
19 $\quad$ **else**
20 $\quad\quad$ Simulate-Interventions$(i + 1, \boldsymbol{\alpha}, x_1, ..., x_n)$ ;
21 $\quad\quad$ **for** *value $v$* **do**
22 $\quad\quad\quad$ Let $x'_1, ..., x'_n := x_1, ..., x_n$;
23 $\quad\quad\quad$ $x'_i := v$;
24 $\quad\quad\quad$ **if** $v \neq x_i$ **then**
25 $\quad\quad\quad\quad$ guess new values $x'_{i+1}, ..., x'_n$
26 $\quad\quad\quad$ **end**
27 $\quad\quad\quad$ Simulate-Interventions$(i + 1, \boldsymbol{\alpha} \cup \{i\}, x'_1, ..., x'_n)$ ;
28 $\quad\quad$ **end**
29 $\quad$ **end**

**Algorithm 1:** Solving $\text{SAT}_{\mathcal{L}_2}^{lin\langle\Sigma\rangle}$

small model property[8] follows that there are only polynomial many, rational probabilities $p_{\mathbf{u}}$. These can be guessed non-deterministically[9].

Next, we combine the terms $\sum_{\mathbf{u}} p_{\mathbf{u}}$ of all sums (implicitly). For example, two sums $\sum_{\mathbf{y}_i} \mathbb{P}([\alpha_i]\delta_i) + \sum_{\mathbf{y}_j} \mathbb{P}([\alpha_j]\delta_j)$ can be rewritten as

$$\sum_{\mathbf{y}_i} \mathbb{P}([\alpha_i]\delta_i) + \sum_{\mathbf{y}_j} \mathbb{P}([\alpha_j]\delta_j) = \sum_{\mathbf{u}} p_{\mathbf{u}} \sum_{\mathbf{y}_i:\mathcal{F}, \mathbf{u}\models[\alpha_i]\delta_i} 1 + \sum_{\mathbf{u}} p_{\mathbf{u}} \sum_{\mathbf{y}_j:\mathcal{F}, \mathbf{u}\models[\alpha_j]\delta_j} 1$$

$$= \sum_{\mathbf{u}} p_{\mathbf{u}} \left( \sum_{\mathbf{y}_i:\mathcal{F}, \mathbf{u}\models[\alpha_i]\delta_i} 1 + \sum_{\mathbf{y}_j:\mathcal{F}, \mathbf{u}\models[\alpha_j]\delta_j} 1 \right).$$

---

[8]Small model property means that whenever there is a satisfying probability distribution, then there is also one with only polynomially many positive elementary probabilities. Note that, e.g., the probability distribution on $n$ binary random variables has $2^n$ entries. Some of the previous results we refer to, e.g., Fagin et al. (1990); Ibeling & Icard (2020), are based on this small property.

[9]The algorithm just guesses them without doing any rewriting of the equations. This paragraph just explains why this guessing is possible.

The algorithm performs this sum over $\mathbf{u}$ in line 5 and calculates the next sums in the subfunction. We can and will ignore the actual values $\mathbf{u}$. Relevant is only that the value of the sums is multiplied by $p_{\mathbf{u}}$ and that the functions $\mathcal{F}$ might change in each iteration.

Recall that the functions $\mathcal{F} = (F_1, \ldots, F_n)$ determine the values of the endogenous variables, that is the value of variable $X_i$ is given by $x_i = F_i(\mathbf{u}, x_1, \ldots, x_{i-1})$. Thereby the functions (i.e. variables) have a causal order $X_1 \prec \ldots \prec X_n$, such that the value $x_i$ only depends on variables $X_j$ with $j < i$. In reverse, this means that each intervention on a variable $X_i$ can only change variables $X_j$ with $j > i$.

Rather than storing the functions $\mathcal{F} = (F_1, \ldots, F_n)$, the algorithm only stores the values $x_i = F_i(\mathbf{u}, x_1, \ldots, x_{i-1})$, i.e. the values of the endogenous variables. The algorithm knows these values as well as the causal order, since they can be guessed non-deterministically.

The subfunction Simulate-Interventions then performs all possible interventions recursively, intervening first on variable $X_1$, then $X_2$, ..., until $X_n$. The parameter $i$ means an intervention on variable $X_i$, $\alpha$ is the set of all previous interventions, and $x_1, \ldots, x_n$ the current values.

In line 20, it proceeds to the next variable, without changing the current variable $X_i$ (simulating all possible interventions includes intervening on only a subset of variables). In line 21, it enumerates all values $x_i'$ for variable $X_i$. If $x_i = x_i'$, then the intervention does nothing. If $x_i \neq x_i'$, then the intervention might change all variables $X_j$ with $j > i$ (because the function $F_j$ might depend on $X_i$ and change its value). This is simulated by guessing the new values $x_j'$. Thereby, we get the new values without considering the functions.

In the last call, line 27, it has completed a set of interventions $\alpha$. The function then searches every occurrence of the interventions $\alpha$ in the (implicitly expanded) input formula. That is, for each sum $\sum_{\mathbf{y}_j : \mathcal{F}, \mathbf{u} \models [\alpha_j] \delta_j} 1$, it counts how often $\alpha_j = \alpha$ occurs in the sum while the values $x_i$ satisfy $\delta_j$. [10]

Since each intervention is enumerated only once, in the end, it obtains for all sums their exact value. It can then verify whether the values satisfy the (in-)equalities of the input.

If a satisfying model exists, the algorithm confirms it, since it can guess the probabilities and the values of the functions. In reverse, if the algorithm returns true, a satisfying model can be constructed. The probabilities directly give a probability distribution $P(\mathbf{u})$. The functions $F_i$ can be constructed because, for each set of values $\mathbf{u}, x_1, \ldots, x_{i-1}$, only a single value for $x_i$ is guessed, which becomes the value of the function.

Algorithm 1 runs in non-deterministic polynomial space and thus in PSPACE. □

### A.3 PROOFS OF SECTION 3.3

*Proof of Theorem 9.* The problem can be solved in NEXP because expanding all sums of a $\text{SAT}_{\mathcal{L}_3}^{lin\langle\Sigma\rangle}$ instance creates a $\text{SAT}_{\mathcal{L}_3}^{lin}$ instance of exponential size, which can be solved non-deterministically in a time polynomial to the expanded size as shown by Mossé et al. (2022).

To prove hardness, we will reduce the satisfiability of a Schönfinkel-Bernays sentence to the satisfiability of $\text{SAT}_{\mathcal{L}_3}^{base\langle\Sigma\rangle}$. The class of Schönfinkel–Bernays sentences (also called Effectively Propositional Logic, EPR) is a fragment of first-order logic formulas where satisfiability is decidable. Each sentence in the class is of the form $\exists\mathbf{x}\forall\mathbf{y}\psi$ whereby $\psi$ can contain logical operations $\wedge, \vee, \neg$, variables $\mathbf{x}$ and $\mathbf{y}$, equalities, and relations $R_i(\mathbf{x}, \mathbf{y})$ which depend on a set of variables, but $\psi$ cannot contain any quantifier or functions. Determining whether a Schönfinkel-Bernays sentence is satisfiable is an NEXP-complete problem Lewis (1980) even if all variables are restricted to binary values Achilleos (2015).

We will represent Boolean values as the value of the random variables, with $0$ meaning FALSE and $1$ meaning TRUE. We will assume that $c = 2$, so that all random variables are binary, i.e. $Val = \{0, 1\}$.

In the proof, we will write (in)equalities between random variables as $=$ and $\neq$. In the binary setting, $X = Y$ is an abbreviation for $(X = 0 \wedge Y = 0) \vee (X = 1 \wedge Y = 1)$, and $X \neq Y$ an abbreviation

---

[10] Rather than incrementing the counter $c_j$ by 1 and then multiplying the final result by $p_{\mathbf{u}}$, we increment it directly by $p_{\mathbf{u}}$.

for $\neg(X = Y)$. To abbreviate interventions, we will write $[w]$ for $[W = w]$, $[\mathbf{w}]$ for interventions on multiple variables $[\mathbf{W} = \mathbf{w}]$, and $[\mathbf{v} \setminus \mathbf{w}]$ for interventions on all endogenous variables except $\mathbf{W}$.

We use random variables $\mathbf{X} = \{X_1, \dots X_n\}$ and $\mathbf{Y} = \{Y_1, \dots Y_n\}$ for the quantified Boolean variables $\mathbf{x}, \mathbf{y}$ in the sentence $\exists \mathbf{x} \forall \mathbf{y} \psi$. For each distinct $k$-ary relation $R_i(z_1, \dots, z_k)$ in the formula, we define a random variable $R_i$ and variables $Z_i^1, \dots, Z_i^k$ for the arguments. For the $j$-th occurrence of that relation $R_i(t_{ij}^1, \dots, t_{ij}^k)$ with $t_{ij}^l \in \{x_1, \dots, x_n, y_1, \dots, y_n\}$, we define another random variable $R_i^j$.

We use the following constraint to ensure that $R_i$ only depends on its arguments:

$$\sum_{\mathbf{v}} \mathbb{P}([z_i^1, \dots, z_i^k]R_i \neq [\mathbf{v} \setminus r_i]R_i) = 0 \tag{6}$$

Thereby $\sum_{\mathbf{v}}$ refers to summing over all values of all endogenous variables[11] in the model and the constraint says that an intervention on $Z_i^1, \dots, Z_i^k$ gives the same result for $R_i$ as an intervention on $Z_i^1, \dots, Z_i^k$ and the remaining variables, excluding $R_i$.

We use the following constraint to ensure that $R_i^j$ only depends on its arguments:

$$\sum_{\mathbf{v}} \mathbb{P}([t_{ij}^1, \dots, t_{ij}^k]R_i^j \neq [\mathbf{v} \setminus r_i^j]R_i^j) = 0 \tag{7}$$

and that $R_i^j$ and $R_i$ have an equal value for equal arguments:

$$\sum_{t_{ij}^1, \dots, t_{ij}^k} \mathbb{P}([T_{ij}^1{=}t_{ij}^1, \dots, T_{ij}^k{=}t_{ij}^k]R_i^j \neq [Z_i^1{=}t_{ij}^1, \dots, Z_i^k{=}t_{ij}^k]R_i) = 0. \tag{8}$$

We add the following constraint for each $X_i$ to ensure that the values of $\mathbf{X}$ are not affected by the values of $\mathbf{Y}$:

$$\sum_{\mathbf{v}} \sum_{\mathbf{y}'} \mathbb{P}([\mathbf{v} \setminus x_i]X_i \neq [\mathbf{v} \setminus (x_i, \mathbf{y}), \mathbf{Y}{=}\mathbf{y}']X_i) = 0 \tag{9}$$

Here the first sum sums over all values $\mathbf{v}$ of all endogenous variables $\mathbf{V}$ (including $X_i$ and $\mathbf{Y}$), and the second sums sums over values for variables $\mathbf{Y}$. The intervention $[\mathbf{v} \setminus x_i]$ intervenes on all variables except $X_i$ and sets the values $\mathbf{y}$ to the values of the first sum. The intervention $[\mathbf{v} \setminus (x_i, \mathbf{y}), \mathbf{Y} = \mathbf{y}']$ intervenes on all variables except $X_i$ and sets the values $\mathbf{y}$ to the values $\mathbf{y}'$ of the second sum. The constraint thus ensures that the value of $X_i$ does not change when changing $\mathbf{Y}$ from $\mathbf{y}$ to $\mathbf{y}'$.

Let $\psi'$ be obtained from $\psi$ by replacing equality and relations on the Boolean values with the corresponding definitions for the random variables:

$$\sum_{\mathbf{y}} \mathbb{P}([\mathbf{y}]\psi') = 2^n \tag{10}$$

Suppose the $\text{SAT}_{\mathcal{L}_3}^{base\langle \Sigma \rangle}$ instance is satisfied by a model $\mathfrak{M}$. We need to show $\exists \mathbf{x} \forall \mathbf{y} \psi$ is satisfiable. Each probability $\mathbb{P}(\dots)$ implicitly sums over all possible values $\mathbf{u}$ of the exogenous variables. The values $\mathbf{x}$ of the variables $\mathbf{X}$ might change together with the values $\mathbf{u}$, however, any values $\mathbf{x}$ that are taken at least once can be used to satisfy $\exists \mathbf{x} \forall \mathbf{y} \psi$: If there was any $\mathbf{x}$ that would not satisfy $\psi$ for all values of $\mathbf{y}$, $\mathbb{P}([\mathbf{y}]\psi')$ would be less than 1 for these values of $\mathbf{x}$ (determined by $\mathbf{u}$) and $\mathbf{y}$, and equation (10) would not be satisfied.

For each relation $R_i$, we choose the values given by the random variable $R_i$. Each occurrence $R_i^j(t_{ij}^1, \dots, t_{ij}^k)$ has a value that is given by $[Z_i^1 = t_{ij}^1, \dots, Z_i^k = t_{ij}^k]R_i$ in the model $\mathfrak{M}$. Due to equations (8) and equations (7), that is the same value as $[T_i^1 = t_{ij}^1, \dots, T_{ij}^k = t_{ij}^k]R_i^j$, which is the value used in $[\mathbf{y}]\psi'$. Since $[\mathbf{y}]\psi'$ is satisfied, so is $\psi$ and $\exists \mathbf{x} \forall \mathbf{y} \psi$.

---

[11] All variables include variables $Z_i^1, \dots, Z_i^k$.

Suppose $\exists\mathbf{x}\forall\mathbf{y}\psi$ is satisfiable. We create a deterministic model $\mathfrak{M}$ as follows: The value of random variables $\mathbf{X}$ is set to the values chosen by $\exists\mathbf{x}$. The relation random variables $R_i$ are functions depending on random variables $Z_i^1, \ldots, Z_i^k$ that return the value of the relation $R_i(z_1, \ldots, z_k)$. The relation random variables $R_i^j$ on arguments $T_{ij}^1, \ldots, T_{ij}^k$ return the value of the relation $R_i(t_{ij}^1, \ldots, t_{ij}^k)$. This satisfies Equation 6 and 7 because the functions only depend on their arguments, and Equation 8 because the functions result from the same relation (so the functions are dependent, but causally independent, which yields a non-faithful model. But the equations do not test for faithfulness or dependences). All other random variables can be kept constant, which satisfies Equation 9 (despite being constant in the model, the causal interventions can still change their values). Finally, Equation 10 holds, because $\psi$ is satisfied for all $\mathbf{Y}$. $\qquad\square$

## A.4 Proofs of Section 4

Mossé et al. (2022) already prove $\mathrm{SAT}_{\mathcal{L}_3}^{poly} \in \exists\mathbb{R}$ when the $\mathcal{L}_3^{poly}$-formula is allowed to neither contain subtractions nor conditional probabilities. We slightly strengthen this result to allow both of them.

**Lemma 15.** $\mathrm{SAT}_{\mathcal{L}_3}^{poly} \in \exists\mathbb{R}$. *This also holds true if we allow the basic terms to contain conditional probabilities.*

*Proof.* (Mossé et al., 2022) show that $\mathrm{SAT}_{\mathcal{L}_3}^{poly}$ without subtraction or conditional probabilities is in $\exists\mathbb{R}$. Their algorithm is given in the form of a NP-reduction from $\mathrm{SAT}_{\mathcal{L}_3}^{poly}$ to ETR and using the closure of $\exists\mathbb{R}$ under NP-reductions. In particular given a $\mathcal{L}_3^{poly}$-formula $\varphi$, they replace each event $\mathbb{P}(\epsilon)$ by the sum $\sum_{\delta\in\Delta^+:\delta\models\epsilon}\mathbb{P}(\delta)$ where $\Delta^+ \subseteq \mathcal{E}_{counterfact}$ is a subset of size at most $|\varphi|$. They add the constraint $\sum_{\delta\in\Delta^+}\mathbb{P}(\delta) = 1$ and then replace each of the $\mathbb{P}(\delta)$ by a variable constrained to be between 0 and 1 to obtain a ETR-formula. Note that the final ETR-formula allows for subtraction, so $\varphi$ is allowed to have subtractions as well. Remains to show how to deal with conditional probabilities. We define conditional probabilities $\mathbb{P}(\delta|\delta')$ to be undefined if $\mathbb{P}(\delta') = 0$ (this proof works similarly for other definitions). In $\varphi$ replace $\mathbb{P}(\delta|\delta')$ by $\frac{\mathbb{P}(\delta,\delta')}{\mathbb{P}(\delta')}$. The resulting ETR-formula then contains some divisions. To remove some division $\frac{\alpha}{\beta}$, we use Tseitin's trick and replace $\frac{\alpha}{\beta}$ by a fresh variable $z$. We then add the constraints $\alpha = z \cdot \beta$ and $\beta \neq 0$ to the formula. $\qquad\square$

Now we are ready to show that $\mathrm{SAT}_{\mathcal{L}_3}^{poly\langle\Sigma\rangle}$ can be solved in NEXP over the Reals.

*Proof of Theorem 10.* By Lemma 15 we know that $\mathrm{SAT}_{\mathcal{L}_3}^{poly}$ is in $\exists\mathbb{R}$. Thus, by (Erickson et al., 2022), there exists an $\mathrm{NP}_{real}$ algorithm, call it $A$, which for a given $\mathcal{L}_3^{poly}$-formula decides if is satisfiable.

To solve the $\mathrm{SAT}_{\mathcal{L}_3}^{poly\langle\Sigma\rangle}$ problem, a $\mathrm{NEXP}_{real}$ algorithm expands firstly all sums of a given instance and creates an equivalent $\mathrm{SAT}_{\mathcal{L}_3}^{poly}$ instance of size bounded exponentially in the size of the initial input formula. Next, the algorithm $A$ is used to decide if the expanded instance is satisfiable. $\qquad\square$

## B    LEVELS OF PEARL'S CAUSAL HIERARCHY: AN EXAMPLE

To illustrate the main ideas behind the causality notions, we present in this section an example that, we hope, will make it easier to understand the formal definitions. In the example, we consider a (hypothetical) scenario involving three attributes represented by binary random variables: pneumonia modeled by $Z = 1$, drug treatment (e.g., with antibiotics) represented by $X = 1$, and recovery, with $Y = 1$ (and $Y = 0$ meaning mortality). Below we describe an SCM which models an unobserved true mechanism behind this setting and the canonical patterns of reasoning that can be expressed at appropriate layers of the hierarchy.

**Structural Causal Model** An SCM is defined as a tuple $(\mathcal{F}, P, \mathbf{U}, \mathbf{X})$ which is of *unobserved nature* from the perspective of a researcher who studies the scenario. The SCM models the ground truth for the distribution $P(\mathbf{U})$ of the population and the mechanism $\mathcal{F}$. In our example, the model assumes three independent binary random variables $\mathbf{U} = \{U_1, U_2, U_3\}$, with probabilities: $P(U_1{=}1) = 0.75, P(U_2{=}1) = 0.8, P(U_3{=}1) = 0.4$, and specifies the mechanism $\mathcal{F} = \{F_1, F_2, F_3\}$ for the evaluation of the three endogenous (observed) random variables $Z, X, Y$ as follows:

| $U_1$ $U_2$ $U_3$ | $P(\mathbf{u})$ | $Z$ $X$ $Y$ |
|---|---|---|
| 0   0   0 | 0.03 | 1   0   0 |
| 0   0   1 | 0.02 | 1   0   1 |
| 0   1   0 | 0.12 | 1   0   0 |
| 0   1   1 | 0.08 | 1   0   1 |
| 1   0   0 | 0.09 | 0   0   0 |
| 1   0   1 | 0.06 | 0   0   1 |
| 1   1   0 | 0.36 | 0   1   0 |
| 1   1   1 | 0.24 | 0   1   1 |

$Z := F_1(U_1) = 1{-}U_1; X := F_2(Z, U_2) = (1{-}Z)U_2; Y := F_3(X, U_1, U_3) = X(1{-}U_1)(1{-}U_3) + (1 - X)(1 - U_1)U_3 + U_1U_3$. Thus, our model determines the distribution $P(\mathbf{u})$, for $\mathbf{u} = (u_1, u_2, u_3)$, and the values for the observed variables, as can be seen above.

The unobserved random variable $U_1$ models all circumstances that lead to pneumonia and $Z$ is a function of $U_1$ (which may be more complex in real scenarios). Getting a treatment depends on having pneumonia but also on other circumstances, like having similar symptoms due to other diseases, and this is modeled by $U_2$. So $X$ is a function of $Z$ and $U_2$. Finally, mortality depends on all circumstances that lead to pneumonia, getting the treatment, and on further circumstances like having other diseases, which are modeled by $U_3$. So $Y$ is a function of $U_1$, $X$, and $U_3$. We always assume that the dependency graph of the SCM is acyclic. This property is also called *semi-Markovian*.

**Layer 1** Empirical sciences rely heavily on the use of observed data, which are typically represented as probability distributions over observed (measurable) variables. In our example, this is the distribution $P$ over $Z, X$, and $Y$. The remaining variables $U_1, U_2, U_3$, as well as the mechanism $\mathcal{F}$, are of unobserved nature. Thus, in our scenario, a researcher gets the probabilities (shown to the right) $P(z, x, y) = \sum_{\mathbf{u}} \delta_{\mathcal{F}, \mathbf{u}}(z, x, y) \cdot P(\mathbf{u})$, where vectors $\mathbf{u} = (u_1, u_2, u_3) \in \{0, 1\}^3$ and $\delta_{\mathcal{F}, \mathbf{u}}(z, x, y) = 1$ if

| $Z$ $X$ $Y$ | $P(z, x, y)$ |
|---|---|
| 0   0   0 | 0.09 |
| 0   0   1 | 0.06 |
| 0   1   0 | 0.36 |
| 0   1   1 | 0.24 |
| 1   0   0 | 0.15 |
| 1   0   1 | 0.10 |

$F_1(u_1){=}z, F_2(z, u_2){=}x$, and $F_3(x, u_1, u_2){=}y$; otherwise $\delta_{\mathcal{F}, \mathbf{u}}(z, x, y) = 0$. The relevant query in our scenario $P(Y{=}1|X{=}1)$ can be evaluated as $P(Y{=}1|X{=}1) = P(Y{=}1, X{=}1)/P(X{=}1) = 0.24/0.6 = 0.4$ which says that the probability for recovery ($Y{=}1$) is only 40% given that the patient took the drug ($X{=}1$). On the other hand, the query for $X{=}0$ can be evaluated as $P(Y{=}1|X{=}0) = P(Y{=}1, X{=}0)/P(X{=}0) = 0.16/0.4 = 0.4$ which may lead to the (wrong, see the next layer) opinion that the drug is irrelevant to recovery.

**Layer 2** Consider a randomized drug trial in which each patient receives treatment, denoted as $do(X{=}1)$, regardless of pneumonia ($Z$) and other conditions ($U_2$). We model this by performing a hypothetical intervention in which we replace in $\mathcal{F}$ the mechanism $F_2(Z, U_2)$ by the constant function 1 and leaving the remaining functions unchanged.

| $U_1$ $U_2$ $U_3$ | $P(\mathbf{u})$ | $Z$ $X{=}1$ $Y$ |
|---|---|---|
| 0   0   0 | 0.03 | 1   1   1 |
| 0   0   1 | 0.02 | 1   1   0 |
| 0   1   0 | 0.12 | 1   1   1 |
| 0   1   1 | 0.08 | 1   1   0 |
| 1   0   0 | 0.09 | 0   1   0 |
| 1   0   1 | 0.06 | 0   1   1 |
| 1   1   0 | 0.36 | 0   1   0 |
| 1   1   1 | 0.24 | 0   1   1 |

| $Z$ $Y$ | $P([X{=}1]z, y)$ |
|---|---|
| 0   0 | 0.45 |
| 0   1 | 0.30 |
| 1   0 | 0.10 |
| 1   1 | 0.15 |

If $\mathcal{F}_{X=1} = \{F_1'{=}F_1, F_2'{=}1, F_3'{=}F_3\}$ denotes the new mechanism, then the *post-interventional* distribution $P([X{=}1]Z, Y)$ is specified as $P([X{=}1]z, y) = \sum_{\mathbf{u}} \delta_{\mathcal{F}_{X=1}, \mathbf{u}}(z, y) \cdot P(\mathbf{u})$, where $\delta_{\mathcal{F}_{X=1}, \mathbf{u}}$ denotes function $\delta$ as above, but for the new mechanism $\mathcal{F}_{X=1}$ (the distribution is shown on the right-hand side). A common and popular notation for the post-interventional probability is $P(Z, Y|do(X{=}1))$. In this paper, we use the notation $P([X{=}1]Z, Y)$ since it is more convenient

for analyses involving counterfactuals. To determine the causal effect of the drug on recovery, we compute, in an analogous way, the distribution $P([X{=}0]Z, Y)$ after the intervention $do(X{=}0)$, which means that all patients receive placebo. Then, comparing the value $P([X{=}1]Y{=}1) = 0.45$ with $P([X{=}0]Y{=}1) = 0.40$, we can conclude that $P([X{=}1]Y{=}1) - P([X{=}0]Y{=}1) > 0$. This can be interpreted as a positive (average) effect of the drug in the population which is in opposite to what has been inferred using the purely probabilistic reasoning of Layer 1. Note that it is not obvious how to compute the post-interventional distributions from the observed probability $P(Z, X, Y)$; Indeed, this is a challenging task in the field of causality.

**Layer 3** The key phenomena that can be modeled and analyzed at this level are counterfactual situations. Imagine, e.g., in our scenario there is a group of patients who did not receive the treatment and died ($X{=}0, Y{=}0$). One may ask, what would be the outcome $Y$ had they been given the treatment ($X{=}1$). In particular, one can ask what is the probability of recovery if we had given the treatment to the patients of this group. Using the formalism of Layer 3, we can express this as a counterfactual query: $P([X{=}1]Y{=}1|X{=}0, Y{=}0) = P([X{=}1](Y{=}1) \wedge (X{=}0, Y{=}0))/P(X{=}0, Y{=}0)$. Note that the event $[X{=}1](Y{=}1) \wedge (X{=}0, Y{=}0)$ incorporates simultaneously two counterfactual mechanisms: $\mathcal{F}_{X=1}$ and $\mathcal{F}$. This is the key difference to Layer 2, where we can only have one. We define the probability in this situation as follows:

$$P([X{=}x](Z{=}z, Y{=}y) \wedge (Z{=}z', X{=}x', Y{=}y')) = \sum_{\mathbf{u}} \delta_{\mathcal{F}_{X=x},\mathbf{u}}(z, y) \cdot \delta_{\mathcal{F},\mathbf{u}}(z', x', y') \cdot P(\mathbf{u}).$$

$X{=}0, Y{=}0$ is satisfied only for $(U_1, U_2, U_3) \in \{(0, 0, 0), (0, 1, 0), (1, 0, 0)\}$ (first table), and of them only $\{(0, 0, 0), (0, 1, 0)\}$ satisfies $[X{=}1]Y{=}1$ (third table). Thus, by marginalizing $Z$, we get $P([X{=}1]Y{=}1|X{=}0, Y{=}0) = 0.15/0.24 = 0.625$ which may be interpreted that more than $62\%$ of patients who did not receive treatment and died would have survived with treatment. Finally, we would like to note that, in general, the events of Layer 3 can be quite involved and incorporate simultaneously many counterfactual worlds.

**Graph Structure of an SCM** Below we remind, how an SCM can be represented in the form of a Directed Acyclic Graph (DAG) and show such a DAG for the model discussed above.

Let $\mathfrak{M} = (\mathcal{F} = \{F_1, \ldots, F_n\}, P, \mathbf{U}, \mathbf{X} = \{X_1, \ldots, X_n\})$ be an SCM. We assume that the model is *Markovian*, i.e. that the exogenous arguments $U_i, U_j$ of $F_i$, resp. $F_j$ are independent whenever $i \neq j$. These exogenous arguments are not shown in the DAG. We note that a general model as discussed above, called *semi-Markovian*, which allows for the sharing of exogenous arguments and allows for arbitrary dependencies among the exogenous variables, can be reduced in a standard way to the Markovian model by introducing auxiliary "unobserved" variables. Thus, in our example, to get a Markovian model, we can assume, $\mathbf{X} = \{X, Y, Z, U_1\}$, where $X, Y, Z$ remain observed variables but $U_1$ is of unobserved nature.

We define that a DAG $\mathcal{G} = (\mathbf{X}, E)$ represents the graph structure of $\mathfrak{M}$ if, for every $X_j$ appearing as an argument of $F_i$, $X_j \to X_i$ is an edge in $E$. DAG $\mathcal{G}$ is called the *causal diagram* of the model $\mathfrak{M}$ Pearl (2009); Bareinboim et al. (2022). The DAG for the discussed SCM therefore has the following form (here, as is usually done in Markovian models, variables $U_2$ and $U_3$ that only affect $X$, respectively $Y$, are omitted):

$$
\begin{array}{ccc}
Z & \longleftarrow & U_1 \\
\downarrow & & \downarrow \\
X & \longrightarrow & Y
\end{array}
$$

Moreover, the DAG of the intervention model discussed in subsection Layer 2, with functional mechanism $\mathcal{F}_{X=1}$ has the following form, meaning that all in-going edges to $X$ are removed from the pre-interventional model:

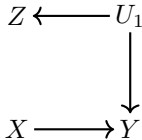

## C  SYNTAX AND SEMANTICS OF THE LANGUAGES OF THE PCH: FORMAL DEFINITIONS

We always consider discrete distributions in the probabilistic and causal languages studied in this paper. We represent the values of the random variables as $Val = \{0, 1, ..., c - 1\}$ and denote by $\mathbf{X}$ the set of random variables used in a system. By capital letters $X_1, X_2, ...$, we denote the individual variables and assume, w.l.o.g., that they all share the same domain $Val$. A value of $X_i$ is often denoted by $x_i$ or a natural number. In this section, we describe syntax and semantics of the languages starting with probabilistic ones and then we provide extensions to the causal systems.

By an *atomic* event, we mean an event of the form $X = x$, where $X$ is a random variable and $x$ is a value in the domain of $X$. The language $\mathcal{E}_{prop}$ of propositional formulas over atomic events is defined as the closure of such events under the Boolean operators $\wedge$ and $\neg$. To specify the syntax of interventional and counterfactual events we define the intervention and extend the syntax of $\mathcal{E}_{prop}$ to $\mathcal{E}_{post\text{-}int}$ and $\mathcal{E}_{counterfact}$, respectively, using the following grammars:

$$
\begin{aligned}
\mathcal{E}_{prop} \text{ is defined by} \quad & \mathbf{p} \quad ::= \quad X = x \mid \neg\mathbf{p} \mid \mathbf{p} \wedge \mathbf{p} \\
\mathcal{E}_{int} \text{ is defined by} \quad & \mathbf{i} \quad ::= \quad \top \mid X = x \mid \mathbf{i} \wedge \mathbf{i} \\
\mathcal{E}_{post\text{-}int} \text{ is defined by} \quad & \mathbf{p_i} \quad ::= \quad [\,\mathbf{i}\,]\,\mathbf{p} \\
\mathcal{E}_{counterfact} \text{ is defined by} \quad & \mathbf{c} \quad ::= \quad \mathbf{p_i} \mid \neg\mathbf{c} \mid \mathbf{c} \wedge \mathbf{c}.
\end{aligned}
$$

Note that since $\top$ means that no intervention has been applied, we can assume that $\mathcal{E}_{prop} \subseteq \mathcal{E}_{post\text{-}int}$.

The PCH consists of three languages $\mathcal{L}_1, \mathcal{L}_2, \mathcal{L}_3$, each of which is based on terms of the form $\mathbb{P}(\delta)$. For the (observational or associational) language $\mathcal{L}_1$, we have $\delta \in \mathcal{E}_{prop}$, for the (interventional) language $\mathcal{L}_2$, we have $\delta \in \mathcal{E}_{post\text{-}int}$ and for the (counterfactual) language $\mathcal{L}_3$, $\delta \in \mathcal{E}_{counterfact}$. The expressive power and computational complexity properties of the languages depend largely on the operations that we are allowed to apply to the basic terms. Allowing gradually more complex operators, we describe the languages which are the subject of our studies below. We start with the description of the languages $\mathcal{T}_i^*$ of terms, with $i = 1, 2, 3$, using the following grammars[12]

$$
\begin{array}{ll|ll}
\mathcal{T}_i^{base} & \mathbf{t} ::= \mathbb{P}(\delta_i) & \mathcal{T}_i^{base\langle\Sigma\rangle} & \mathbf{t} ::= \mathbb{P}(\delta_i) \mid \sum_x \mathbf{t} \\
\mathcal{T}_i^{lin} & \mathbf{t} ::= \mathbb{P}(\delta_i) \mid \mathbf{t} + \mathbf{t} & \mathcal{T}_i^{lin\langle\Sigma\rangle} & \mathbf{t} ::= \mathbb{P}(\delta_i) \mid \mathbf{t} + \mathbf{t} \mid \sum_x \mathbf{t} \\
\mathcal{T}_i^{poly} & \mathbf{t} ::= \mathbb{P}(\delta_i) \mid \mathbf{t} + \mathbf{t} \mid -\mathbf{t} \mid \mathbf{t} \cdot \mathbf{t} & \mathcal{T}_i^{poly\langle\Sigma\rangle} & \mathbf{t} ::= \mathbb{P}(\delta_i) \mid \mathbf{t} + \mathbf{t} \mid -\mathbf{t} \mid \mathbf{t} \cdot \mathbf{t} \mid \sum_x \mathbf{t}
\end{array}
$$

where $\delta_1$ are formulas in $\mathcal{E}_{prop}$, $\delta_2 \in \mathcal{E}_{post\text{-}int}$, $\delta_3 \in \mathcal{E}_{counterfact}$.

The probabilities of the form $\mathbb{P}(\delta_i)$ are called *primitives* or *basic terms*. In the summation operator $\sum_x$, we have a dummy variable $x$ which ranges over all values $0, 1, \ldots, c - 1$. The summation $\sum_x \mathbf{t}$ is a purely syntactical concept which represents the sum $\mathbf{t}[0/x] + \mathbf{t}[1/x] + \ldots + \mathbf{t}[c-1/x]$, where by $\mathbf{t}[v/x]$, we mean the expression in which all occurrences of $x$ replaced with value $v$. For example, for $Val = \{0, 1\}$, the expression $\sum_x \mathbb{P}(Y=1, X=x)$ semantically represents $\mathbb{P}(Y=1, X=0) + \mathbb{P}(Y=1, X=1)$. We note that the dummy variable $x$ is not a (random) variable in the usual sense and that its scope is defined in the standard way.

In the table above, the terms in $\mathcal{T}_i^{base}$ are just basic probabilities with the events given by the corresponding languages $\mathcal{E}_{prop}, \mathcal{E}_{post\text{-}int}$, or $\mathcal{E}_{counterfact}$. Next, we extend terms by being able to compute sums of probabilities and by adding the same term several times, we also allow for weighted sums with weights given in unary. Note that this is enough to state all our hardness results. All matching upper bounds also work when we allow for explicit weights given in binary. In the case of $\mathcal{T}_i^{poly}$, we are allowed to build polynomial terms in the primitives. On the right-hand side of the table, we have the same three kinds of terms, but to each of them, we add a marginalization operator as a building block.

The polynomial calculus $\mathcal{T}_i^{poly}$ was originally introduced by Fagin, Halpern, and Megiddo (Fagin et al., 1990) (for $i = 1$) to be able to express conditional probabilities by clearing denominators. While this works for $\mathcal{T}_i^{poly}$, this does not work in the case of $\mathcal{T}_i^{poly\langle\Sigma\rangle}$, since clearing denominators

---

[12]In the given grammars we omit the brackets for readability, but we assume that they can be used in a standard way.

with exponential sums creates expressions that are too large. But we could introduce basic terms of the form $\mathbb{P}(\delta_i|\delta)$ with $\delta \in \mathcal{E}_{prop}$ explicitly. All our hardness proofs work without conditional probabilities but all our matching upper bounds are still true with explicit conditional probabilities. Expression as $\mathbb{P}(X{=}1) + \mathbb{P}(Y{=}2) \cdot \mathbb{P}(Y{=}3)$ is a valid term in $\mathcal{T}_1^{poly}$ and $\sum_z \mathbb{P}([X{=}0](Y{=}1, Z{=}z))$ and $\sum_z \mathbb{P}(([X{=}0]Y{=}1), Z{=}z)$ are valid terms in the language $\mathcal{T}_3^{poly\langle\Sigma\rangle}$, for example.

Now, let $Lab = \{\text{base}, \text{base}\langle\Sigma\rangle, \text{lin}, \text{lin}\langle\Sigma\rangle, \text{poly}, \text{poly}\langle\Sigma\rangle\}$ denote the labels of all variants of languages. Then for each $* \in Lab$ and $i = 1, 2, 3$ we define the languages $\mathcal{L}_i^*$ of Boolean combinations of inequalities in a standard way:

$$\mathcal{L}_i^* \text{ is defined by } \quad \mathbf{f} ::= \mathbf{t} \leq \mathbf{t}' \mid \neg\mathbf{f} \mid \mathbf{f} \wedge \mathbf{f}, \quad \text{where } \mathbf{t}, \mathbf{t}' \text{ are terms in } \mathcal{T}_i^*.$$

Although the language and its operations can appear rather restricted, all the usual elements of probabilistic and causal formulas can be encoded. Namely, equality is encoded as greater-or-equal in both directions, e.g. $\mathbb{P}(x) = \mathbb{P}(y)$ means $\mathbb{P}(x) \geq \mathbb{P}(y) \wedge \mathbb{P}(y) \geq \mathbb{P}(x)$. The number 0 can be encoded as an inconsistent probability, i.e., $\mathbb{P}(X{=}1 \wedge X{=}2)$. In a language allowing addition and multiplication, any positive integer can be easily encoded from the fact $\mathbb{P}(\top) \equiv 1$, e.g. $4 \equiv (1+1)(1+1) \equiv (\mathbb{P}(\top) + \mathbb{P}(\top))(\mathbb{P}(\top) + \mathbb{P}(\top))$. If a language does not allow multiplication, one can show that the encoding is still possible. Note that these encodings barely change the size of the expressions, so allowing or disallowing these additional operators does not affect any complexity results involving these expressions.

To define the semantics of the languages, we use a structural causal model (SCM) as in (Pearl, 2009, Sec. 3.2). An SCM is a tuple $\mathfrak{M} = (\mathcal{F}, P, \mathbf{U}, \mathbf{X})$, such that $\mathbf{V} = \mathbf{U} \cup \mathbf{X}$ is a set of variables partitioned into exogenous (unobserved) variables $\mathbf{U} = \{U_1, U_2, ...\}$ and endogenous variables $\mathbf{X}$. The tuple $\mathcal{F} = \{F_1, ..., F_n\}$ consists of functions such that function $F_i$ calculates the value of variable $X_i$ from the values $(\mathbf{x}, \mathbf{u})$ of other variables in $\mathbf{V}$ as $F_i(\mathbf{pa}_i, \mathbf{u}_i)$ [13], where $\mathbf{Pa}_i \subseteq \mathbf{X}$ and $\mathbf{U}_i \subseteq \mathbf{U}$. $P$ specifies a probability distribution of all exogenous variables $\mathbf{U}$. Since variables $\mathbf{X}$ depend deterministically on the exogenous variables via functions $F_i$, $\mathcal{F}$ and $P$ obviously define the joint probability distribution of $\mathbf{X}$. Throughout this paper, we assume that domains of endogenous variables $\mathbf{X}$ are discrete and finite. In this setting, exogenous variables $\mathbf{U}$ could take values in any domains, including infinite and continuous ones. A recent paper (Zhang et al., 2022) shows, however, that any SCM over discrete endogenous variables is equivalent for evaluating post-interventional probabilities to an SCM where all exogenous variables are discrete with finite domains. As a consequence, throughout this paper, we assume that domains of exogenous variables $\mathbf{U}$ are discrete and finite, too.

For any basic $\mathcal{E}_{int}$-formula $X_i{=}x_i$ (which, in our notation, means $do(X_i{=}x_i)$), we denote by $\mathcal{F}_{X_i=x_i}$ the function obtained from $\mathcal{F}$ by replacing $F_i$ with the constant function $F_i(\mathbf{v}) := x_i$. We generalize this definition for any interventions specified by $\alpha \in \mathcal{E}_{int}$ in a natural way and denote as $\mathcal{F}_\alpha$ the resulting functions. For any $\varphi \in \mathcal{E}_{prop}$, we write $\mathcal{F}, \mathbf{u} \models \varphi$ if $\varphi$ is satisfied for values of $\mathbf{X}$ calculated from the values $\mathbf{u}$. For $\alpha \in \mathcal{E}_{int}$, we write $\mathcal{F}, \mathbf{u} \models [\alpha]\varphi$ if $\mathcal{F}_\alpha, \mathbf{u} \models \varphi$. And for all $\psi, \psi_1, \psi_2 \in \mathcal{E}_{counterfact}$, we write $(i)$ $\mathcal{F}, \mathbf{u} \models \neg\psi$ if $\mathcal{F}, \mathbf{u} \not\models \psi$ and $(ii)$ $\mathcal{F}, \mathbf{u} \models \psi_1 \wedge \psi_2$ if $\mathcal{F}, \mathbf{u} \models \psi_1$ and $\mathcal{F}, \mathbf{u} \models \psi_2$. Finally, for $\psi \in \mathcal{E}_{counterfact}$, let $S_{\mathfrak{M}} = \{\mathbf{u} \mid \mathcal{F}, \mathbf{u} \models \psi\}$. We define $[\![\mathbf{e}]\!]_{\mathfrak{M}}$, for some expression $\mathbf{e}$, recursively in a natural way, starting with basic terms as follows $[\![\mathbb{P}(\psi)]\!]_{\mathfrak{M}} = \sum_{\mathbf{u} \in S_{\mathfrak{M}}(\psi)} P(\mathbf{u})$ and, for $\delta \in \mathcal{E}_{prop}$, $[\![\mathbb{P}(\psi|\delta)]\!]_{\mathfrak{M}} = [\![\mathbb{P}(\psi \wedge \delta)]\!]_{\mathfrak{M}} / [\![\mathbb{P}(\delta)]\!]_{\mathfrak{M}}$, assuming that the expression is undefined if $[\![\mathbb{P}(\delta)]\!]_{\mathfrak{M}} = 0$. For two expressions $\mathbf{e}_1$ and $\mathbf{e}_2$, we define $\mathfrak{M} \models \mathbf{e}_1 \leq \mathbf{e}_2$, if and only if, $[\![\mathbf{e}_1]\!]_{\mathfrak{M}} \leq [\![\mathbf{e}_2]\!]_{\mathfrak{M}}$. The semantics for negation and conjunction are defined in the usual way, giving the semantics for $\mathfrak{M} \models \varphi$ for any formula $\varphi$ in $\mathcal{L}_3^*$.

---

[13] We consider recursive models, that is, we assume the endogenous variables are ordered such that variable $X_i$ (i.e. function $F_i$) is not affected by any $X_j$ with $j > i$.

