# OpenReview forum: "From Probability to Counterfactuals: the Increasing Complexity of Satisfiability in Pearl's Causal Hierarchy"
_ICLR.cc/2025/Conference — ICLR 2025 Poster_

### Official Review · Reviewer_K9we · 2024-11-03

**Soundness:** 3
**Presentation:** 3
**Contribution:** 4
**Rating:** 8
**Confidence:** 3

**Summary:**

This paper studies the complexity of satisfiability in three increasingly complex probabilistic frameworks: purely probabilistic, interventional, and counterfactional. They build on previous work which does not allow for marginalisation, and the authors add results that allow for marginalisation.
Their findings are interesting: for each of the three frameworks, they find to which complexity class it belongs, and interestingly the three classes differ (under widely accepted assumptions).
The authors motive their approach well, and cite the relevant literature, at least so far as I can see. The paper gives proofs for all the claimes in an appendix, so the findings are well supported.
This paper adds to the field of causality in probability theory, which is becoming increasingly more relevant, with new results that have the potential to influence further studies in this field.

**Strengths:**

In my opinion, this paper studies a very relevant question, which naturally arise. The current interest in causality in probability theory makes this paper even more relevant. The results obtained are important on its own right, and may well influence further research.

This paper is well written. The authors took great care in explaining the concepts, and generally succeeded with this (but not everywhere in their paper – see my "questions" section).

One particular thing I appreciated, was the use of 'proof ideas'. In my opinion, this is great to convey the idea, at the same time reducing the risk that the reader loses themselves in details. This seems a nice way for conferences, to my taste.

In all, I am very positive about this paper. I identify later a small number of points where minor improvements are possible. Note that this field is not my expertise, so while I am rather confident that this is a good paper, I am open to arguments against this.

**Weaknesses:**

In general, I am positive about this paper, and I only identify a small number of weaknesses.

While the authors generally motivate their approach, and introduce the necessary concepts, well, at some points there is improvement possible.
- For instance, in Table 1 the authors should make more clear that the languages L1, L2 and L3 refer to their introduction on the next page in Section 2.1. The authors do not refer there to L1, L2, and L3, which makes it more harder to make this link.
- An instance where the motivation could be improved, is Section 3, the sentences following that of Equation (3). There, the authors remark that adding marginalisation to a language does not change the expressivenes of it. It comes to wonder, then, why the authors are particularly interested in the questions they answer: if adding marginalisation does not change the expressivenes, why is it important to add this? The authors expand a bit on this in the Introduction, but I would rephrase the sentence here, or expand on it, in order to make the paper feel better motivated.
Also, how do we know that adding marginalisation to a language does not change the expressivenes of it? It may be a well known fact, but I would add a reference.

I end this section with a small list of small remarks and typos:
- Abstract, last sentence: Please be more specific about which open problem in which field. Ideally, give a reference that states that the problem you describe is open.
- Page 2, line 065: add a space between "see, e.g.," and "(Pearl, 1988;".
- Page 6, line 314:  I would write that the sum "ranges" over all values, instead of "sums" over it.

**Questions:**

Here are three questions that would increase the readibility in my opinion:
1. Page 4, line 182: The basic term P( [ X_1=x_1] X_2=x_2 ) does not belong to T_1^base, if I understand it correctly. The way it's written now, seems to suggest that it does. Please clarify this.
2. Page 6, line 289: What does it mean for a sum to be "equivalent" to another sum? Does the author(s) mean "equal"?
3. Theorem 9: The proof idea only seems to (informally, but I like this here, as I mentioned in the "strengths" section) show that the satisfiability problems are NEXP-hard instead of NEXP-complete. The proof in the appendix does, however, prove the desired result. I suggest that the authors rephrase this, or (if I'm mistaken here) explain why the current proof idea (informally) shows that they are NEXP-complete.

---

> ### Author Response · Authors · 2024-11-21
>
> Thanks for your positive assessment of our work and the suggestions to improve the presentation.  We will incorporate them.
>
> Regarding the importance of marginalization, it is the standard notation in probability theory, any textbook on probability theory would include marginalization operators. It is more surprising that logicians have removed the marginalization operators from their considerations.  Without marginalization, the expressiveness is the same for finite domains by expanding the sum, e.g. ∑_{x=1}^5 P(X=x) becomes  P(X=1)+P(X=2)+P(X=3)+P(X=4)+P(X=5). However, when studying algorithmic and complexity aspects of probabilistic, causal, and counterfactual reasoning, enabling the marginalization for representing queries in a common, compact ways plays a crucial role, because the computational complexity is a function of the length of the input query. We can try to emphasizes this more in the introduction.
>
> Questions:
> ----
>
> - Page 4: That is true. We will clarify this.
> - Page 6: Yes, we mean equal. We will clarify this, too.
> - Theorem 9: Thanks for pointing this out. We will add a sentence about the containment to the proof idea.

---

> > ### Comment · Reviewer_K9we · 2024-11-25
> > **Reply to the authors**
> >
> > I'd like to thank the authors for their replies.
> >
> > With my comment about marginalisation, I meant to indicate that I think the authors, in their paper, should motivate better why marginalisation is important, especially given their remark that it does not change the expressiveness of the language. I am convinced that it is, indeed, an important concept.
> >
> > I believed that this is a good paper, and I still believe this.

---

### Official Review · Reviewer_J9DB · 2024-11-03

**Soundness:** 3
**Presentation:** 3
**Contribution:** 3
**Rating:** 5
**Confidence:** 4

**Summary:**

The paper studies the complexity of the satisfiability problem of various logics for describing properties of structural causal models. The logics incorporate probability comparisons of statements including interventional counterfactuals. The authors give a very rigorous study on the complexity of the satisfiability problem of fragments of the logics they consider.

**Strengths:**

The paper is very rigorous, and the results are surely interesting for the community of researcher working on logics for causal reasoning.

**Weaknesses:**

While the call for papers mentions causal reasoning as one of it's topics knowing, I do not think that a paper whose main contributions are pure complexity results for logics for causal reasoning is suitable for ICLR. Though the precise complexity results obtained might have some impact on the algorithmic aspects of causal reasoning. The paper is an almost pure logic paper studying computational complexity landscape of the logic they consider and as such is a valuable contribution that should be carefully reviewed and published in a more suitable venue. I would urge the authors to resubmit the paper to a venue that can better appreciate the results There are numerous conferences of this kind, including A* conferences such as LICS and KR, but also JELIA, CSL, and many other.

**Questions:**

None.

---

> ### Author Response · Authors · 2024-11-21
>
> We thank the reviewer for their review, but regret that they offer such a low score whilst hardly engaging with our paper at all.
>
> We do not agree with the reviewer, that our paper is a pure logic paper or that it is a pure complexity theory paper for logics for causal reasoning.
>
> Our main contribution is to establish the possibilities and limitations of algorithmic reasoning across the three levels of Pearl's hierarchy, including probabilistic, causal, and counterfactual reasoning. The algorithmic aspects of these reasoning tasks are critical in the fields of ML and AI and have been the focus of extensive research within the community. This includes both the development of specific algorithms and heuristics for inference and the study of the computational efficiency limitations of these methods. The rich body of literature on these topics is exemplified by the references provided in our paper, including the seminal work by Cooper (1990) and by Fagin et al. (1990) as well as by Ibeling and Icard (2020), Littman et al. (1998), Nilsson (1986), Park and Darwiche (2004), Roth (1996), Shpitser and Pearl (2008), van der Zander et al. (2023), published, e.g., in Artificial Intelligence, Journal of Artificial Intelligence Research, IJCAI, or AAAI.
>
> The importance of establishing exact completeness results for probabilistic, causal, and counterfactual reasoning, as stated in our paper, lies in their implications for algorithmic approaches to solving these problems. Assuming the widely accepted belief that ∃R ≠ NP or NP ≠ PSPACE, the completeness of a problem highlights inherent limitations in applying algorithmic techniques, such as dynamic programming, divide-and-conquer, tree-width algorithms, SAT or ILP solvers, which are effective for NP-complete problems. This, in turn, justifies the use of heuristics, despite their potential for exponential worst-case complexity.
>
> While the algorithmic and complexity aspects of probabilistic reasoning are well-studied and well-understood (e.g., Fagin et al. (1990), Cooper (1990), Ibeling and Icard (2020)), we have extended these results to queries expressed in a commonly used format. Furthermore, we have provided the first complexity results for answering causal and counterfactual queries formulated in a widely accepted manner. In our opinion, these results constitute an important contribution to the field of “causal reasoning” listed as one of the ICLR topics.
>
> On the other hand, our work is NOT purely a logic paper. As emphasized in the paper, all languages that use the summation operator are semantically equivalent to those without it. What is novel about using the summation operator, however, is its ability to provide a compact and common representation of queries. This representation is crucial for studying the algorithmic and complexity aspects of probabilistic, causal, and counterfactual reasoning.

---

> > ### Comment · Reviewer_J9DB · 2024-11-24
> > **Reply to the comment**
> >
> > Thank you for your reply. I will marginally increase my score, since like you mention, knowing the precise complexity results might have some impact on the algorithmic aspects of causal reasoning. However, I still maintain my view that a more suitable venue for the work described in the paper is either a journal (as mentioned by another reviewer) or a theoretical computer science conference. I would be in favour in accepting the submission (after some rewriting of course) to essentially any upper mid-tier TCS conference, and consider it to be somewhat out of scope for ICLR.
> >
> > Below some further comments and questions.
> >
> > You mention above and also in line 254 that "As emphasized in the paper, all languages that use the summation operator are semantically equivalent to those without it." Could you please expand on what you mean by having the same expressiveness? In particular, you seem to fix VAL to some finite set. Is this to be understood that all your results (e.g. complexity of SAT) are relative to a fixed finite VAL? Or are you saying that the summation operator can be uniformly eliminated regardless of the cardinality of VAL? The role of VAL in the results should be made more clear in the paper.
> >
> > lines 195-200. The definition of SCMs here is very compact and should be expanded. E.g., you use the notation Pa_i for the parents of X_i (I suppose). A few words explaining the setting and notation used would be helpful.
> >
> > line 206: Typo. "S_M" should be "S_M(\psi)".
> >
> > line 242: It would be good to mention here, or before, where "\exists R" and "succ-\exists R" are situated wrt. other complexity classes.
> >
> > There are some related works that you might be interested in:
> >
> > Several recent papers by Barbero et al. consider various properties (mainly expressivity and axiomatisations) of languages for causality and interventions. https://dblp.org/pid/18/11469.html

---

> > > ### Author Response · Authors · 2024-11-25
> > > **Replay to the comments**
> > >
> > > Thanks for your feedback. We answer your questions or comments below.
> > >
> > > > Could you please expand on what you mean by having the same expressiveness?
> > >
> > > We explain the concept of expressibility in the introduction to Section 3, and already in Section 1 (L: 144-146), we mention the increasing expressibility of languages ​​and provide references to Section 3, where this issue is discussed in more detail.
> > >
> > > Again, for two languages L_a and L_b, language L_a is less expressive than L_b if there exists two SCMs M and M' that are indistinguishable in L_a, but that can be distinguished by some formula Ψ in the more expressive language L_b. This means that (1) for any formula φ in L_a we have  M |= φ  iff  M' |= φ and (2) there exists formula Ψ in L_b such that M |= Ψ  but  not (M' |= Ψ). In the second paragraph of the Section 3 (L: 256-264), we show, for example, that any language of the probabilistic layer is less expressive than L_2^{base} — the basic language of the interventional layer — by constructing the SCMs M and M' and showing that the formula P([X_1=1]X_2=1) = P([X_1=1]X_2=0) satisfies the required properties. As we pointed out in Section 3, the increasing expressiveness of the languages ​​of the PCH has been studied previously in the literature, see e.g. (Pearl, 2009; Bareinboim et al., 2022; Mossé et al., 2022; Suppes & Zanotti, 1981).
> > >
> > > So, admitting summation to a language L does not increase its expressiveness since, for any formula φ' in the language L' with summation, one obtains an equivalent formula φ by expansions the terms with summations, as e.g. ∑_{x=1}^5 P(X=x) becomes P(X=1)+P(X=2)+P(X=3)+P(X=4)+P(X=5).
> > >
> > > At this point, we would like to emphasize again that our results show for the first time a strictly increasing computational complexity of reasoning (for basic and linear languages ​​with summation): from probabilistic to causal to counterfactual reasoning, which coincides with an increasing expressive power of the corresponding languages.
> > >
> > > > In particular, you seem to fix VAL to some finite set. Is this to be understood that all your results (e.g. complexity of SAT) are relative to a fixed finite VAL? Or are you saying that the summation operator can be uniformly eliminated regardless of the cardinality of VAL?
> > >
> > > Yes, we assume the set VAL to be fixed and finite, but arbitrary (of cardinality at least two). That is, our results hold for any set VAL, as long as it is a finite set and contains at least two elements.
> > >
> > > That also means when considering, e.g. satisfiability problems, VAL is not part of the input: In our setting, we assume the input is a formula φ for the given language (with random variables over finite set VAL, as e.g., Boolean values). This is a common approach also used in previous studies, see, for example: (Fagin et al., 1990), (Ibeling & Icard, 2020), (Mossé et al., 2022), (van der Zander et al., 2023). This is justified by the fact that the completeness results are already true for such variants of the problems.
> > >
> > > It would also be justified to only consider the case of VAL={0,1}. Any larger finite set VAL could be reduced to {0,1} using a binary encoding of the values with additional variables. For example, if VAL was {0,1,2}, and we had a formula containing P(X=2), one could replace the variable X with two binary variables X_1 and X_2 and replace P(X=2) by P(X_1=1, X_2=0), which would not change the complexity class.
> > >
> > > > The definition of SCMs here is very compact and should be expanded. E.g., you use the notation Pa_i for the parents of X_i (I suppose). A few words explaining the setting and notation used would be helpful.
> > >
> > > Thanks for the comment. Yes, Pa_i denotes parents of X_i. We will improve the presentation in the final version.
> > >
> > > > line 206: Typo. "S_M" should be "S_M(\psi)".
> > >
> > > Yes, you are right. Sorry for the typo.
> > >
> > > > line 242: It would be good to mention here, or before, where "\exists R" and "succ-\exists R" are situated wrt. other complexity classes.
> > >
> > > Thanks for the comment. We will extend the equation (1) as follows :
> > >
> > > NP ⊆ ∃R,  NP^PP ⊆ PSPACE ⊆ NEXP  ⊆ succ-∃R ⊆ EXPSPACE
> > >
> > > and mention that the relationship between  ∃R and  NP^PP is unknown.
> > >
> > > > There are some related works that you might be interested in: Several recent papers by Barbero et al. consider various properties
> > >
> > > Thanks for the hints to the literature.
> > >
> > > In fact, various aspects related to counterfactual reasoning are currently the subject of intensive research, including algorithmic and complexity issues. One of our main messages is that the computational complexity of counterfactual reasoning (for the most general queries) remains the same as for common probabilistic reasoning. As we noted in our paper, this is quite a surprising result, as the difference between the expressive power of both settings is huge.

---

### Official Review · Reviewer_cUWX · 2024-11-04

**Soundness:** 2
**Presentation:** 2
**Contribution:** 2
**Rating:** 3
**Confidence:** 3

**Summary:**

This paper studies the complexity issues of the satisfiability and validity problems of formulas and shows an increasing complexity within the framework of Pearl’s Causal Hierarchy of reasoning: observational, interventional and counterfactual.

**Strengths:**

(1)The paper studies an important theoretical problem for causal reasoning and answers some open problems in this field.
(2)The paper provides a good motivation and detailed literature.

**Weaknesses:**

(1)The paper is very technical, condensed and very hard to understand. Hence it is difficult to check the mathematical soundness and evaluate the contributions. But the mathematical proofs turns out to more important than anything else including originality and insights in this paper. The main text consists of a list of theoretical results. The topic is also very limited. Since the paper is mainly about the pure complexity results, probably it is more appropriate as a journal submission or a pure theory conference.
(2)Some important results in the paper are related to some uncommon complexity classes like succ-ExistentialR-completeness. The paper fails to provide a clear explanation of this class and some motivation of the relationship.
(3)There is no a concrete example which demonstrates the increasing complexity results. Such an example would help understand the motivation behind the complexity results for different levels of languages.
(4)The introduction in Section 2.1 of Pearl’s PCH is very messy. It would take some hard time to understand the main components and the structure of the hierarchy.  It would be better to illustrate the graph structures of SCMs and give some examples to illustrate the increasing structures.

**Questions:**

(1)There are several operations in the structure: sum, marginalization, conditioning. What are their roles in the increasing complexities of the hierarchy?
(1)Could you give nontrivial examples to demonstrate different levels of reasoning and to illustrate the corresponding increasing complexities?

---

> ### Author Response · Authors · 2024-11-21
>
> Thanks for your review. Below we respond to your comments and answer your questions.
>
> Comments
> ----
>
> (1) We think that checking validity is an important task in ML and AI and understanding how hard it is to check that a given property is valid, e.g., in a Structural Causal Model (SCM) of interest, is an important issue. We would like to emphasize that although the computational complexity results of our work mainly concern satisfiability problems, they also establish validity complexity because satisfiability and validity testing are complementary.
>
> (2) ∃R, and also succ-∃R, are complexity classes that have become very important in the recent years. See the current survey [Schaefer, Cardinal, Miltzow The existential theory of the reals as a complexity class: A compendium. arXiv (2024)]. As we mentioned in our response to the reviewer J9DB, the importance of establishing exact completeness results for probabilistic, causal, and counterfactual reasoning, as stated in our paper, lies in their implications for algorithmic approaches to solving these problems. Assuming widely accepted complexity assumptions, the completeness of a problem highlights inherent limitations in applying algorithmic techniques and justifies the use of approximations and heuristics.
>
> In particular, using succ-∃R-completeness as a yardstick for measuring computational complexity of problems, we show that the complexity of counterfactual reasoning (for the most general queries) remains the same as for common probabilistic reasoning. This is quite a surprising result, as the difference between the expressive power of both settings is huge.
>
> (3) For concrete examples of queries, which demonstrate the increasing complexity results, see our answers to your questions below.
>
> (4) In fact, Pearl's causal hierarchy can be thought of as providing ever more opportunities to manipulate the SCM: from pure observation, through intervention (manipulation) of the SCM, to the simultaneous generation of several manipulated SCMs. We agree that to explain a single intervention, it would indeed be appropriate to use the graph structure of the SCM. However, due to space constraints we could not do this. Instead, to help understand these concepts, we provide in Section B in appendix, an example for reasoning on all levels of the hierarchy. In the final version, we will try to present the concepts already in Section 2.1 in a more clear way and additionally, in Section B we will use the graph structure of the SCM to explain an intervention.
>
> Questions
> ----
>
> (1) "Operations in the structure and their roles in the increasing complexities of the hierarchy": The relation is summarized in Table 1. There are three dimensions of increasing difficulty:
>
> First, the three levels of Pearls hierarchy (from left to right) which allow to formulate basic terms P(δ) used in queries. The complexity increases from: observational (probabilistic) events δ, like X = 1 ∧ Y = 1 (meaning, e.g., drug treatment and recovery), through interventional, like [X = 1] Y = 1 (meaning, e.g., recovery after intervention "drug treatment"), to the counterfactual events δ, like [X=1](Y =1) | (X=0, Y =0)) (meaning, e.g., a conditional event: recovery after drug treatment conditioned by an event, not treatment and the patient died).
> (Note, that we use the notation $P([X=x] Y=y)$ for the post-interventional distribution, which is often denoted as P(Y=y | do(X=x)), as the notation is more convenient for counterfactual reasoning)
>
> Second, the power of arithmetic operations used to formulate queries involving the basic terms P(δ): none = "basic" (one can only formulate queries using relations like P(δ) ≤ P(δ') and combine them using Boolean operators ∧ and ¬), linear operations demoted as "lin" (one can formulate queries like P(δ) + P(δ') ≤ P(δ'')), addition and multiplication = "poly" (one can formulate queries like P(δ)*P(δ') + P(δ'') ≤ 1/2). Here the computational complexity increases from basic, through lin, to poly.
>
> The third dimension is whether we do not allow marginalization (upper three lines of the table) or we have marginalization (lower three lines), as e.g. in $\sum_{x,y} (P(X=x, Y=y) - P(X=x) P(Y=y))^2=0$ to express independence of X and Y.
>
> The case without marginalization was settled in previous work and our main contribution is to settle the computational complexity with marginalization. (Exponential) sums are the concrete realization of marginalization. In this sense, sums and marginalization are the same. Conditioning has only minor effects on the hardness of the problem.
>
> (2) The examples from left to right are given by the power of Pearl's hierarchy. The difference from basic/lin to poly is due to the fact that one can use multiplications to formulate statements about reals numbers. An example at the interventional level involving marginalization is the adjustment formula. If you want to see the increasing complexities, one needs to look at the formulas in our hardness proofs.

---

> > ### Author Response · Authors · 2024-11-29
> > **Your comments**
> >
> > Dear Reviewer cUWX,
> >
> > thank you again for your comments and valuable feedback. We would like to know if our responses addressed your concerns and questions? If there are any remaining issues, we are happy to address them as soon as possible.
> >
> > Many thanks!
> >
> > Your sincerely,
> >
> > Authors of submission 11078

---

### Official Review · Reviewer_enRi · 2024-11-05

**Soundness:** 3
**Presentation:** 3
**Contribution:** 3
**Rating:** 8
**Confidence:** 3

**Summary:**

The paper studies the computational complexity of the satisfiability problems of queries under different causal layers. In particular, it considers queries that may involve marginalizations ($\sum$) and proves that the sat problems with basic and linear terms under L1, L2, L3 layers belong to the complexity classes of $NP^{PP}$, PSPACE, and NEXP. Moreover, it completes the open problem in Van der Zander et al. to show that the sat problem with polynomial term under L3 also belongs to succ-$\exists R$.

**Strengths:**

- I checked the proofs and most of them make sense to me. The proofs are also quite rigorous in my opinion.
- The paper is succinct and clearly structured.
- The results are novel and are important contributions to the complexity of casual satisfiability problems.

**Weaknesses:**

I think the main challenge is for the readers to process the complexity results.

Here are some suggestions:
1. I think providing more background on the complexity classes in the preliminaries section can be very helpful. Currently the paper points to [Arora & Barak 2009] textbook for more background about the complexity classes. If the page limit permits, it's better to summarize the definitions for each complexity class considered in this paper along with some complete problems in each class. This should make the paper more self-contained.
2. A lot of contents in Appendix C should be moved to Section 2.1. For example, I got lost in the definitions of base, lin, and poly, etc.. Also, please provide some concrete examples so that we know immediately their differences.
3. I think it is worth stressing that the satisfiability problem does not assume anything about SCMs, nor does the causal structure. In the classical work on the complexity of probabilistic inference, it is often assumed that both the structure and parameterizations are known (Remark 5 in the paper). In causal effect identification, the causal structure (graph) is often assumed to be known.

**Questions:**

1. Are there any (potential) applications of the satisfiability problems mentioned in this work? For example, do they have any implications in causal identifiability or causal discovery (structure learning)? I can see the applications of the validity problems but not sure about the satisfiability problems.
2. Why/when do queries with marginalization become important? How does this relate to existing works on marginalizing (forgetting) variables from causal models?
3. In Appendix A.2 proof of Lemma 14: "A small model property follows that there are only polynomial many, rational probabilities $p_u$" I don’t know what "small model property" is.

Minor typos:
- Line 816: "the second sums" -> "the second sum"
- Line 863: "$\frac{P(\delta)}{P(\delta')}$" -> "$\frac{P(\delta)}{P(\delta, \delta')}$", maybe?

---

> ### Author Response · Authors · 2024-11-21
>
> - Thanks for your thoughtful suggestions on the structure of the paper. We will try to incorporate them.
> - In particular we will emphasize more clearly that the satisfiability problem (and its complement, the validity problem) does not assume anything about SCMs, nor does the causal structure. However, we will additionally mention that our languages allow queries of the form φ ⇒ Ψ which enable us to verify satisfiability, resp., the validity, for the formula Ψ in Structural Causal Models (SCMs) which satisfy properties expressed by the formula φ, whereby φ can encode a graph structure. Thus, the formalism used in our work allows for the formulation of a wide range of queries.
> - In addition, we will provide examples of queries expressed in the analyzed languages across the PCH.
>
> Questions
>
> - Indeed, validity testing is one of the main and natural applications. Since satisfiability and validity testing are complements of each other, it does not matter which of the problems one studies. From a technical point of view, satisfiability testing problems are often easier to study. Below we discuss some examples of validation testing applications where queries are typically expressed in languages that use the marginalization / summation operator.
> - A famous example for a formula with marginalization is the adjustment-set formula
>  $P([x] y) = \sum_{z} P{y | x, z) P(z)
> (AS), see e.g. [Pearl, Glymour, Jewell: Causal Inference in Statistics - A Primer, Eq. (3.5)]. Again, we would like to remind that in our work we use the notation $P([x] y)$ for the post-interventional distribution, which is commonly denoted as $P(y | do(x))$. However, for the analysis of counterfactual reasoning, the notation $P([x] y)$ is much more convenient.
> - A validity query regarding the adjustment formula can be expressed as  φ ⇒ (AS), where φ is a formula describing certain properties that a SCM of interest should satisfy. In particular, φ can specify the causal structure (graph) of the SCM. This example nicely illustrates the application of validity testing to identify causal effects.
> - Another example would be the prominent front-door adjustment [Pearl, Biometrika, 82(4):669–688, 1995] formula
>  $P([x] y) = \sum_{z} P(z | x} \sum_{x'} P{y | x', z) P(x')$ (FD). Again, we can formulate a query as φ ⇒ (FD) and test the validity of this implication. Similarly, validity testing can be applied in other cases.
> - Marginalization for forgetting variables is a special case when the sum of a single probabilistic primitive is taken, where the primitive only contains a conjunction of variables. We consider this case in Proposition 1. If the sum is taken of a more complex term like in the adjustment formula or a primitive containing disjunctions, the result is more complex than just forgetting variables.
> - Small model property means that whenever there is a satisfying probability distribution, then there is also one with only polynomially many non-zero elementary probabilities. (Note that the probability distribution on n random variables has 2^n entries.) Some of the referenced literature has based their results on such a small model property.

---

> > ### Comment · Reviewer_enRi · 2024-11-27
> >
> > Thank you for the clarifications and detailed examples. I've raised my rating.

---

### Author Response · Authors · 2024-11-28
**Revised PDF**

We would like to thank all the reviewers for their valuable time and thoughtful reviews. We also thank them for their constructive comments to improve the quality of our paper, which we have tried to take into account in the revised version. We have attempted to do the revision while staying within the permitted page limit. Below we provide more details on the modifications made.

## Reviewer enRi:

> Stressing that the satisfiability problem does not assume anything about SCMs.

done: see the 2-nd Paragraph in Sec.2.2

> In Appendix A.2 proof of Lemma 14. I don’t know what "small model property" is.

done

> Minor typos

done


## Reviewer cUWX:

>  Some important results in the paper are related to some uncommon complexity classes like succ-ExistentialR-completeness

In Sec. 2.2 we give a new reference to a current survey paper Schaefer et al. (2024) on ∃R, including its succinct version — the complexity class succ∃R. Moreover, we give a new paragraph (the last one) in Subsection "Our Contribution" in the Introduction. There we stress the significance of this class and the importance of our completeness results for this class as well as for the other classes shown in equation (1) in the new version.

>  There is no a concrete example which demonstrates the increasing complexity results

We now show in the Introduction the relations between complexity classes for which completeness results have been proven (new equation (1)). Under the commonly accepted complexity assumptions, all inclusion relations are strict, which implies an increasing complexity of satisfiability problems (and their complements, the validity problems) across the PCH.

> Illustrate the graph structures of SCMs

To help understand the basic concepts, including interventions, we provide additional material on SCM graph structures in Section B of the Appendix.


## Reviewer J9DB:

> In particular, you seem to fix VAL to some finite set. Is this to be understood that all your results (e.g. complexity of SAT) are relative to a fixed finite VAL?

We added a sentence mentioning that it is a fixed set.

> lines 195-200. The definition of SCMs here is very compact and should be expanded. E.g., you use the notation Pa_i for the parents of X_i (I suppose). A few words explaining the setting and notation used would be helpful.

We added some sentences explaining the parents, the relationship between endogenous and exogenous variables, and that this definition is not based on graphs

> line 206: Typo. "S_M" should be "S_M(\psi)".

done

> line 242: It would be good to mention here, or before, where "\exists R" and "succ-\exists R" are situated wrt. other complexity classes.

done: see Introduction new Eq. (1)


## Reviewer K9we:

> Page 4, line 182: The basic term P( [ X_1=x_1] X_2=x_2 ) does not belong to T_1^base, if I understand it correctly. The way it's written now, seems to suggest that it does. Please clarify this.

done

> 2. Page 6, line 289: What does it mean for a sum to be "equivalent" to another sum? Does the author(s) mean "equal"?

done

> 3. Theorem 9: The proof

We added a sentence about the containment

---

### Meta-Review · Area_Chair_rM9A · 2024-12-20

**Metareview:**

Pearl's Causal Hierarchy (PCH) outlines three levels of reasoning—probabilistic, interventional, and counterfactual—each representing increasing complexity in understanding causation. This paper explores the computational complexity of satisfiability problems within PCH, focusing on probabilistic and causal languages. Specifically, it examines whether a given set of formulas in these languages can be satisfied by some model. The main contribution is determining the exact computational complexities: languages that include addition and marginalization (via summation) result in NP^PP-, PSPACE-, and NEXP-complete satisfiability problems corresponding to the probabilistic, causal, and counterfactual levels of PCH, respectively. These findings show a clear increase in complexity across the PCH levels. Furthermore, for full languages that also allow multiplication, the complexity of satisfiability at the counterfactual level remains consistent with the lower levels, thereby resolving an open problem previously posed by Van der Zander et al. The paper also conducts a thorough analysis of the complexity of satisfiability across various logic fragments used to describe structural causal models, including probability comparisons and interventional counterfactuals. Overall, the study significantly enhances the understanding of the computational challenges inherent in different layers of causal reasoning.

All the reviewers comment that the paper provides a solid and rigorous theoretical contribution to causal reasoning. The reviewers provide a list of suggestions for improving the quality of the presentation. I recommend this paper for publication and invite the authors to consider the reviewers' suggestions for preparing the final draft of the manuscript.

**Additional Comments On Reviewer Discussion:**

In the discussion period, the authors have addressed some of the questions raised by the reviewers to clarify the scope and interpretation of the results. Almost all reviewers engaged in the review process.

---

### Decision · Program_Chairs · 2025-01-22

Accept (Poster)